# Improve Agents without Retraining: Parallel Tree Search with Off-Policy Correction

**Assaf Hallak** *
NVIDIA Research
ahallak@nvidia.com

**Gal Dalal** *
NVIDIA Research
gdalal@nvidia.com

**Steven Dalton**
NVIDIA Research
sdalton@nvidia.com

**Iuri Frosio**
NVIDIA Research
ifrosio@nvidia.com

**Shie Mannor**
NVIDIA Research
smannor@nvidia.com

**Gal Chechik**
NVIDIA Research
gchechik@nvidia.com

## Abstract

Tree Search (TS) is crucial to some of the most influential successes in reinforcement learning. Here, we tackle two major challenges with TS that limit its usability: *distribution shift* and *scalability*. We first discover and analyze a counter-intuitive phenomenon: action selection through TS and a pre-trained value function often leads to lower performance compared to the original pre-trained agent, even when having access to the exact state and reward in future steps. We show this is due to a distribution shift to areas where value estimates are highly inaccurate and analyze this effect using Extreme Value theory. To overcome this problem, we introduce a novel off-policy correction term that accounts for the mismatch between the pre-trained value and its corresponding TS policy by penalizing under-sampled trajectories. We prove that our correction eliminates the above mismatch and bound the probability of sub-optimal action selection. Our correction significantly improves pre-trained Rainbow agents without any further training, often more than doubling their scores on Atari games. Next, we address the scalability issue given by the computational complexity of exhaustive TS that scales exponentially with the tree depth. We introduce Batch-BFS: a GPU breadth-first search that advances all nodes in each depth of the tree simultaneously. Batch-BFS reduces runtime by two orders of magnitude and, beyond inference, enables also training with TS of depths that were not feasible before. We train DQN agents from scratch using TS and show improvement in several Atari games compared to both the original DQN and the more advanced Rainbow.

## 1 Introduction

Tree search (TS) is a fundamental component of Reinforcement Learning (RL) [46] used in some of the most successful RL systems [42, 44]. For instance, Monte-Carlo TS (MCTS) [10] achieved superhuman performance in board games like Go [44], Chess [45], and Bridge [5]. MCTS gradually unfolds the tree by adding nodes and visitation counts online and storing them in memory for future traversals. This paradigm is suitable for discrete state-spaces where counts are aggregated across

---

*Equal contribution (random order)

35th Conference on Neural Information Processing Systems (NeurIPS 2021).

multiple iterations as the tree is built node-by-node, but less suitable for continuous state-spaces or image-based domains like robotics and autonomous driving [1]. For the same reason, MCTS cannot be applied to improve pre-trained agents without collecting their visitation statistics in training iterations.

Instead, in this work, we conduct the TS "on-demand" by expanding the tree up to a given depth at each state. Our approach handles continuous and large state-spaces like images without requiring any memorization. This on-demand TS can be performed both at training or inference time. Here, we focus our attention on the second case, which leads to score improvement without any re-training. This allows one to better utilize existing pre-trained agents even without having the ability or resources to train them. For example, a single AlphaGo training run is estimated to cost 35 million USD [1]. In other cases, even when compute budget is not a limitation, the setup itself makes training inaccessible. For example, when models are distributed to end-clients with too few computational resources to train agents in their local custom environments.

We run TS for inference as follows. For action selection, we feed the states at the leaves of the spanned tree to the pre-trained value function. We then choose the action at the root according to the branch with the highest discounted sum of rewards and value at the leaves. Our approach instantly improves the scores of agents that were already trained for long periods (see Sec. 5.1). Often, such improvement is possible because the value function is not fully realizable with a function approximator, e.g., a deep neural network; TS can then overcome the limitation of the model. In practice, TS requires access to a forward model that is fed with actions to advance states and produce rewards. Here, we build on the recently published CuLE [11] – an Atari emulator that runs on GPU. This allows us to isolate the fundamental properties of TS without the added noise of learned models such as those described in Sec.6.

Performing TS on-demand has many benefits, but it also faces limitations. We identify and analyze two major obstacles: distribution shift and scalability.

First, we report a counter-intuitive phenomenon when applying TS to pre-trained agents. As TS looks into the future, thus utilizing more information from the environment, one might expect that searching deeper should yield better scores. Surprisingly, we find that in many cases, the opposite happens: action selection based on vanilla TS can drastically impair performance. We show that performance deteriorates due to a distribution shift from the original pre-trained policy to its corresponding tree-based policy. We analyze this phenomenon by quantifying the probability of choosing a sub-optimal action when the value function error is high. This occurs because for values of out-of-distribution states, larger variance translates to a larger bias of the maximum. Our analysis leads to a simple, computationally effective off-policy correction term based on the Bellman error. We refer to the resulting TS as the Bellman-Corrected Tree-Search (BCTS) algorithm. BCTS yields monotonically improving scores as the tree depth increases. In several Atari games, BCTS even more than doubles the scores of pre-trained Rainbow agents [22].

The second limitation is scalability: the tree grows exponentially with its depth, making the search process computationally intensive and limiting the horizon of forward-simulation steps. To overcome this limitation, we propose Batch-BFS: a parallel GPU adaptation of Breadth-First Search (BFS), which brings the runtime down to a practical regime. We measured orders-of-magnitude speed-up compared to alternative approaches. Thus, in addition to improving inference, it also enables training tree-based agents in the same order of training time without a tree. By combining Batch-BFS with DQN [32] and training it with multiple depths, we achieve performance comparable or superior to Rainbow – one of the highest-scoring variants of the DQN-based algorithm family.

**Our Contributions.** (1) We identify and analyze a distribution-shift that impairs post-training TS. (2) We introduce a correction mechanism and use it to devise BCTS: an efficient algorithm that improves pre-trained agents, often doubling their scores or more. (3) We create Batch-BFS, an efficient TS on GPU. (4) We use Batch-BFS to train tree-based DQN agents and obtain higher scores than DQN and Rainbow.

---

[1]In the case of continuous state-spaces or image-based domains, MCTS can be used by reconstructing trajectories from action sequences only in deterministic environments. Also, MCTS requires a fixed initial state because the root state has to either exist and be found in the MCTS tree or alternatively a new tree has to be built and reiterated for that initial state.

## 2   Preliminaries

Our framework is an infinite-horizon discounted Markov Decision Process (MDP) [39]. An MDP is defined as the 5-tuple $(\mathcal{S}, \mathcal{A}, P, r, \gamma)$, where $\mathcal{S}$ is a state space, $\mathcal{A}$ is a finite action space, $P(s'|s, a)$ is a transition kernel, $r(s, a)$ is a reward function, $\gamma \in (0, 1)$ is a discount factor. At each step $t = 0, 1, \ldots$, the agent observes the last state $s_t$, performs an action $a_t$ and receives a reward $r_t$. The next state is then sampled by $s_{t+1} \sim P(\cdot|s_t, a_t)$. For brevity, we denote $A := |\mathcal{A}|$.

Let $\pi : \mathcal{S} \to \mathcal{A}$ be a stationary policy. Let $Q^\pi : \mathcal{S} \times \mathcal{A} \to \mathbb{R}$ be the state-action value of a policy $\pi$, defined in state $s$ as $Q^\pi(s, a) \equiv \mathbb{E}^\pi \left[ \sum_{t=0}^\infty \gamma^t r(s_t, \pi(s_t)) \big| s_0 = s, a_0 = a \right]$, where $\mathbb{E}^\pi$ denotes expectation w.r.t. the distribution induced by $\pi$. Our goal is to find a policy $\pi^*$ yielding the optimal value $Q^*$ such that $Q^*(s, a) = \max_\pi r(s, a) + \gamma \mathbb{E}_{s' \sim P(\cdot|s,a)} \max_{a'} Q^\pi(s', a')$. It is well known that

$$Q^*(s, a) = r(s, a) + \gamma \mathbb{E}_{s' \sim P(\cdot|s,a)} \max_{a'} Q^*(s', a'), \qquad \pi^*(s) = \arg\max_a Q^*(s, a).$$

**Vanilla tree search.** To ease notations and make the results concise, we limit the analysis to deterministic transitions[2], i.e., an action sequence $(a_0, \ldots, a_{d-1})$, starting at $s_0$ leads to a corresponding trajectory $(s_0, \ldots, s_d)$. Nonetheless, the results can be extended to a stochastic setup by working with the marginal probability over the trajectory. Then, for a policy $\pi_o$, let the $d$-step Q-function

$$Q_d^{\pi_o}(s, a) = \max_{(a_k)_{k=1}^d \in \mathcal{A}} \left[ \sum_{t=0}^{d-1} \gamma^t r(s_t, a_t) + \gamma^d Q^{\pi_o}(s_d, a_d) \right]_{s_0 = s, a_0 = a}, \tag{1}$$

and similarly let $\hat{Q}_d^{\pi_o}(s, a)$ be the d-step Q-function estimator that uses an estimated Q-function $\hat{Q}^{\pi_o}$ instead of $Q^{\pi_o}$. Finally, denote by $\pi_d$ the $d$-step greedy policy

$$\pi_d(s) := \arg\max_{a \in \mathcal{A}} \hat{Q}_d^{\pi_o}(s, a). \tag{2}$$

## 3   Solving the Tree Search Distribution Shift

In this section, we show how to leverage TS to improve a pre-trained policy. We start by demonstrating an intriguing phenomenon: The quality of agents degrades when using vanilla TS. We then analyze the problem and devise a solution. The core idea of our approach is to distinguish between actions that are truly good, and those that are within the range of noise. By quantifying the noise using the Bellman error and problem parameters, we find the exact debiasing that yields the optimal signal-to-noise separation.

### 3.1   Performance degradation with vanilla tree search

We focus on applying TS at inference time, without learning. We begin with a simple experiment that quantifies the benefit of using TS given a pre-trained policy.

A TS policy has access to future states and rewards and, by definition, is optimal when the tree depth goes to infinity. Hence, intuitively, one may expect a TS policy to improve upon $\pi_o$ for finite depths as well. To test this, we load Rainbow agents $\hat{Q}^{\pi_o}$, pre-trained on 50M frames; they are publicly available in [24] and achieve superhuman scores as in [22]. We use them to test TS policies $\pi_d$ with multiple depths $d$ on several Atari benchmarks. Surprisingly, the results (red curves in Fig. 5, Sec. 5.1) show that TS reduces the total reward, sometimes to scores of random policies, in various games and depths. The drop is particularly severe for TS policies with $d = 1$ — a fact later explained by our analysis in Thm. 3.5.

We find the reason for this performance drop to be the poor generalization of the value function to states outside the stationary $\pi_o$'s distribution. Fig. 1 shows a typical, bad action selection in Atari Breakout by a depth-1 TS. The table on the right reports the estimated Q-values of the root state (first column) and of every state at depth 1 (last four columns). Since the ball is dropping, 'Left' is the optimal action. Indeed, this corresponds with the Q-values at the root. However, at depth 1,

---

[2]The Atari environments we experiment on here are indeed close to deterministic. Their source of randomness is the usage of a random number of initial noop actions [33].

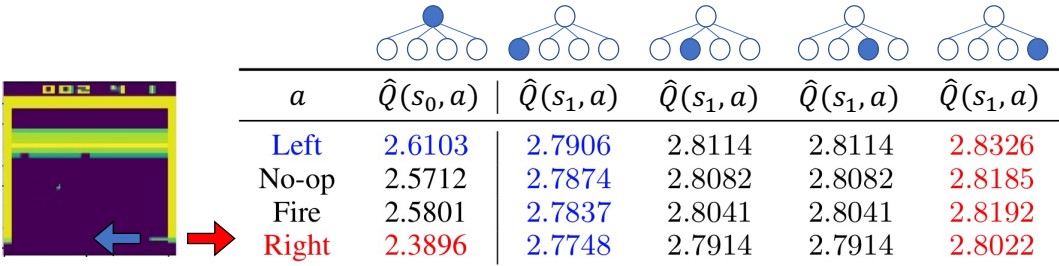

| $a$ | $\hat{Q}(s_0,a)$ | $\hat{Q}(s_1,a)$ | $\hat{Q}(s_1,a)$ | $\hat{Q}(s_1,a)$ | $\hat{Q}(s_1,a)$ |
|---|---|---|---|---|---|
| Left | 2.6103 | 2.7906 | 2.8114 | 2.8114 | 2.8326 |
| No-op | 2.5712 | 2.7874 | 2.8082 | 2.8082 | 2.8185 |
| Fire | 2.5801 | 2.7837 | 2.8041 | 2.8041 | 2.8192 |
| Right | 2.3896 | 2.7748 | 2.7914 | 2.7914 | 2.8022 |

Figure 1: **A failure of vanilla tree search.** *Left*: An Atari Breakout frame. *Right*: Q-values of TS for the frame on the left. Rows correspond to the action taken at the considered depth, which is $d = 0$ for the first column and $d = 1$ for the four others. The action at the root of the tree is color-coded: Red for 'Right', and blue for 'Left'.

the Q-values of the future state that corresponds to choosing 'Left' (second column) are the lowest among all depth-1 future states. Subsequently, the depth-1 TS policy selects 'Right'.

During training, towards convergence, the trained policy mostly selects 'Left' while other actions are rarely sampled. Therefore, expanding the tree at inference time generates states that have been hardly observed during training and are consequently characterized by inaccurate Q-value estimates. In the case of Fig. 1, the Q-value for 'Left' should indeed be low because the agent is about to lose the game. As for the other, poorly sampled states, regression towards a higher mean leads to over-estimated Q-values. Similar poor generalization has been observed in previous studies [16] and is interpreted as an off-policy distribution shift [35].

Beyond the anecdotal example in Fig. 1, additional evidence supports our interpretation regarding the distribution shift. We first consider the error in the value estimate captured by the Bellman error minimized in training. We compare the average Bellman error of the action chosen by $\pi_o$ to all other actions at the tree root. When averaging over 200 episodes, we find that the error for actions chosen by $\pi_o$ is consistently lower than for other actions: $\times 1.5$ lower for Breakout, and $\times 2$ for Frostbite. We also measure the off-policy distribution shift between $\pi_o$ and the TS policy (that utilizes $\pi_o$) by counting disagreements on actions between the two policies. In Breakout, $\pi_o$ and $\pi_1$ agreed only in 18% of the states; in Frostbite, the agreement is only 1.96%. Such a level of disagreement between a well-trained agent and its one-step look-ahead extension is surprising. On the other hand, it accounts for the drastic drop in performance when applying TS, especially in Frostbite (Fig. 5).

## 3.2 Analysis of the degradation

We analyze the decision process of a policy $\pi_d$, given a pre-trained value function estimator, in a probabilistic setting. Our analysis leads to a simple correction term given at the end of the section. We also show how to compute this correction from the TS. Formally, we are given a policy represented as a value function $\hat{Q}^{\pi_o}$, which we feed to the $d$-step greedy policy (2) for each action selection. Policy training is a stochastic process due to random start states, exploration, replay buffer sampling, etc. The value estimator $\hat{Q}^{\pi_o}$ can thus be regarded as a random variable, with an expectation that is a deterministic function $Q^{\pi_o}$ with a corresponding policy $\pi_o$, where 'o' stands for 'original'.

In short, we bound the probability that a sub-optimal action falsely appears more attractive than the optimal one. We thus wish to conclude with high probability whether $a_0 = \arg\max_a \hat{Q}_d^{\pi_o}(s,a)$ is indeed optimal, i.e., $Q_d^{\pi_o}(s,a_0) \geq Q_d^{\pi_o}(s,a) \; \forall a \in \mathcal{A}$.

As we have shown in Sec. 3.1 that states belonging to trajectories that follow $\pi_o$ have lower value estimation noise, we model this effect via the following two assumptions. Here, we denote by $t = 0$ the time when an agent acts and not as the time step of the episode.

**Assumption 1.** *Let* $\sigma_o, \sigma_e \in \mathbb{R}^+$ *s.t.* $0 < \sigma_o < \sigma_e$. *For action sequence* $(a_0, \ldots, a_{d-1})$ *and corresponding state trajectory* $(s_0, \ldots, s_d)$,

$$\hat{Q}^{\pi_o}(s_d, a_d) \sim \begin{cases} \mathcal{N}(Q^{\pi_o}(s_d, a_d), \sigma_o^2) & \text{if } a_0 = \pi_o(s_0) \\ \mathcal{N}(Q^{\pi_o}(s_d, a_d), \sigma_e^2) & \text{otherwise.} \end{cases}$$

Assumption 1 presumes that a different choice of the first action $a_0$ yields a different last state $s_d$. This is commonly the case for environments with large state spaces, especially when stacking observations as done in Atari. While assuming a normal distribution is simplistic, it still captures the essence of the search process. Regarding the expectation, recall that $\pi_o$ is originally obtained from the previous training stage via gradient descent with a symmetric loss function $\mathcal{L}$. Due to the symmetric loss, the estimate $\hat{Q}_\theta^{\pi_o}$ is unbiased, i.e., $\mathbb{E}[\hat{Q}_\theta^{\pi_o}] = Q^{\pi_o}$. Regarding the variance, towards convergence, the loss is computed on replay buffer samples generated according to the stationary distribution of $\pi_o$. The estimate of the value function for states outside the stationary distribution is consequently characterized by a higher variance, i.e. $\sigma_o < \sigma_e$. We also show that this separation of variance occurs in the data, as detailed in the last paragraph of Sec. 3.1.

After conditioning the variance on whether $\pi_o$ was followed, we similarly split the respective sub-trees. To split, we assume the cumulative reward along the tree and values at the leaves depend only on (i) the root state and (ii) whether $\pi_o$ was selected at that state.

**Assumption 2.** *For action sequence $(a_0, \ldots a_{d-1})$ and corresponding trajectory $(s_0, \ldots, s_d)$,*

$$\sum_{t=0}^{d-1} \gamma^t r(s_t, a_t) = \begin{cases} R_o(s_0) \text{ if } a_0 = \pi_o(s_0) \\ R_e(s_0) \text{ otherwise,} \end{cases} \qquad Q^{\pi_o}(s_d, a_d) = \begin{cases} \mu_o(s_0) \text{ if } a_0 = \pi_o(s_0) \\ \mu_e(s_0) \text{ otherwise,} \end{cases}$$

*with $R_o, R_e, \mu_o, \mu_e$ being functions of $s$.*

Assumption 2 could be replaced with a more detailed consideration of each trajectory, but we make it for simplicity. This assumption considers the worst-case scenario: the rewards are unhelpful in determining the optimal policy, and all leaves are equally likely to mislead the $d$-step greedy policy. That is, there is no additional signal to separate between the $A^d$ leaves besides the initial action.

Assuming from now on that Assumptions 1 and 2 hold, we can now explicitly express the distribution of the maximal value among the leaves using Generalized Extreme Value (GEV) theory [9].

**Lemma 3.1.** *The estimates $\hat{Q}_d^{\pi_o}(s, \pi_o(s))$ and $\max_{a \neq \pi_o(s)} \hat{Q}_d^{\pi_o}(s, a)$ are GEV-distributed with parameters given in Appendix A.1.*

All proofs are deferred to Appendix A. Using the GEV distribution, we can now quantify the bias stemming from the maximization in each of two sub-trees corresponding $\pi_o$ vs. all other actions.

**Lemma 3.2.** *It holds that*

$$\mathbb{E}\left[\hat{Q}_d^{\pi_o}(s, \pi_o(s))\right] = Q_d^{\pi_o}(s, \pi_o(s)) + \gamma^d B_o(\sigma_o, A, d)$$

$$\mathbb{E}\left[\max_{a \neq \pi_o(s)} \hat{Q}_d^{\pi_o}(s, a)\right] = \max_{a \neq \pi_o(s)} Q_d^{\pi_o}(s, a) + \gamma^d B_e(\sigma_e, A, d),$$

*where the biases $B_o, B_e$ are given in Appendix A.2, and satisfy $0 \leq B_o(\sigma_o, A, d) < B_e(\sigma_e, A, d)$.*

Lemma 3.2 conveys the main message of our analysis: the variance of terms being maximized translates to a positive shift in the expectation of the maximum. Hence, even if $\mu_o(s) > \mu_e(s)$ for a certain $s$, a different action than $\pi_o(s)$ can be chosen with non-negligible probability as the bias in $\mu_e(s)$ is greater than the one in $\mu_o(s)$. To compensate for this bias that gives an unfair advantage to the noisier nodes of the tree, we introduce a penalty term that precisely cancels it.

### 3.3 BCTS: Bellman Corrected Tree Search

Instead of selecting actions via (2), in BCTS we replace $\hat{Q}_d^{\pi_o}$ with the corrected $Q_d^{\text{BCTS}}$ defined by

$$\hat{Q}_d^{\text{BCTS}, \pi_o}(s, a) := \begin{cases} \hat{Q}_d^{\pi_o}(s, a) & \text{if } a_0 = \pi_o(s_0), \\ \hat{Q}_d^{\pi_o}(s, a) - \gamma^d \left(B_e(\sigma_e, A, d) - B_o(\sigma_o, A, d)\right) & \text{otherwise,} \end{cases} \qquad (3)$$

and we denote the BCTS policy by

$$\pi_d^{\text{BCTS}} := \arg\max_a \hat{Q}_d^{\text{BCTS}, \pi_o}(s, a). \qquad (4)$$

In the following result, we prove that BCTS indeed eliminates undesirable bias.

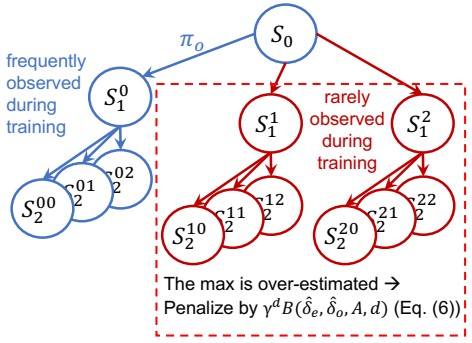

Figure 2: **BCTS Algorithm.** Exploring the full tree reveals out-of-distribution states (red). These were less visited during training and tend to have highly variable scores, leading to high overestimation error. The penalty term $B(\hat{\delta}_e, \hat{\delta}_o, A, d)$ (see (6)) cancels the excess bias. The Bellman errors $\hat{\delta}_e, \hat{\delta}_o$ are extracted from the tree at depth 1.

**Theorem 3.3.** *The relation* $\mathbb{E}\left[\hat{Q}_d^{BCTS,\pi_o}(s, \pi_o(s))\right] > \mathbb{E}\left[\max_{a \neq \pi_o(s)} \hat{Q}_d^{BCTS,\pi_o}(s, a)\right]$ *holds if and only if* $Q_d^{\pi_o}(s, \pi_o(s)) > \max_{a \neq \pi_o(s)} Q_d^{\pi_o}(s, a)$.

The biases $B_o$, $B_e$ include the inverse of the cumulative standard normal distribution $\Phi^{-1}$ (Appendix A.2). We now approximate them with simple closed-form expressions that are highly accurate for $d \geq 2$ (Appendix A.7). These approximations help revealing how the problem parameters dictate prominent quantities such as the correction term in (3) and the probability in Thm. 3.5 below.

**Lemma 3.4.** *When* $A^{d-1} \gg 1$*, the correction term in* (3) *can be approximated with*

$$B_e(\sigma_e, A, d) - B_o(\sigma_o, A, d) \approx \sqrt{2 \log A}\left(\sigma_e \sqrt{d} - \sigma_o \sqrt{d-1}\right) - (\sigma_e - \sigma_o)/2. \qquad (5)$$

The bias gap in (5) depends on the ratio between $\sigma_e$ and $\sigma_o$; this suggests that TS in different environments will be affected differently. As $\sigma_e > \sigma_o$, the bias gap is positive. This is indeed expected, since the maximum over the sub-trees of $a_0 \neq \pi_o(s)$ includes noisier elements than those in the sub-tree of $a_0 = \pi_o(s)$. Also, in (5), $\sigma_e \sqrt{d}$ dominates $\sigma_o \sqrt{d-1}$, making the bias gap grow asymptotically with $\sqrt{d \log A}$. This rate reflects how the number of elements being maximized over affects the bias of their maximum.

Next, we bound the probability of choosing a sub-optimal action when using BCTS. In Appendix A, Thm. A.1, we give an exact bound to that probability without assuming $A^{d-1} \gg 1$. Here, we apply Lemma. 3.4 to give the result in terms of $\sigma_o$, $\sigma_e$.

**Theorem 3.5.** *When* $A^{d-1} \gg 1$*, the policy* $\pi_d^{BCTS}(s)$ *(see* (4)*) chooses a sub-optimal action with probability bounded by:*

$$\Pr\left(\pi_d^{BCTS}(s) \notin \arg\max_a Q_d^{\pi_o}(s, a)\right) \leq \left(1 + \frac{6d \log A\left(Q_d^{\pi_o}(s, \pi_o(s)) - \max_{a \neq \pi_o(s)} Q_d^{\pi_o}(s, a)\right)^2}{\gamma^{2d} \pi^2 \left(\sigma_o^2 + \sigma_e^2\right)}\right)^{-1}.$$

The fraction in the above bound is a signal-to-noise ratio: The expected value difference in the numerator represents the signal, while the variances in the denominator represent the noise. In addition, the signal is "amplified" by $d \log A$ because, after applying the correction from (3), a larger number of tree nodes amount to a more accurate maximum estimator. A similar amplification also occurs due to nodes being deeper and is captured by $\gamma^d$ in the denominator.

The factor $\gamma^d$ also appears in the correction term from (3). This correction is the bias gap from (5), scaled by $\gamma^d$. Hence, asymptotically, while the bias gap grows with $\sqrt{d}$, the exponential term is more dominant; so, overall, this product decays with $d$. This decay reduces the difference between the vanilla and BCTS policies for deeper trees by lowering the portion of the estimated value compared to the exact reward. As we show later, this phenomenon is consistent with our experimental observations.

Computing the correction term requires estimates of $\sigma_o$ and $\sigma_e$. As a surrogate for the error, we use the Bellman error. To justify it, let us treat the values of $\hat{Q}_d^{\pi_o}$ at different depths as samples of $\hat{Q}^{\pi_o}$. Then, the following result holds for its variance estimator. Note that we do not need Assumptions 1 and 2 to prove the following result.

**Algorithm 1** Batch-BFS

**Input:** GPU environment $\mathcal{G}$, value network $Q_\theta$, depth $d$
**Init tensors:** state $\bar{S} = [s_0]$, action $\bar{A} = [0, 1, 2, .., A - 1]$, reward $\bar{R} = [0]$
**for** $i_d = 0$ **to** $d - 1$ **do**
    $\bar{S} \leftarrow \bar{S} \times A, \quad \bar{R} \leftarrow \bar{R} \times A$             // Replicate state and reward tensors $A$ times
    $\bar{r}, \bar{S}' = \mathcal{G}([\bar{S}, \bar{A}])$                       // Feed $[\bar{S}, \bar{A}]$ to simulator and advance
    $\bar{R} \leftarrow \bar{R} + \gamma^{i_d} \bar{r}, \quad \bar{S} \leftarrow \bar{S}'$          // Accumulate discounted reward
    $\bar{A} \leftarrow \bar{A} \times A$                           // Replicate action tensor $A$ times
**end for**
$\bar{R} \leftarrow \bar{R} + \gamma^d \max_a Q_\theta(\bar{S}, a)$          // Accumulate discounted value of states at depth $d$
**Return** $\lfloor (\arg\max \bar{R})/A^{d-1} \rfloor$        // Return optimal action at the root

**Proposition 3.6.** *Let $\widehat{var}_n[X]$ be the variance estimator based on $n$ samples of X. Then,*

$$\widehat{var}_{n=2}[\hat{Q}^{\pi_o}(s,a)] = \left( \hat{Q}_1^{\pi_o}(s,a) - \hat{Q}_0^{\pi_o}(s,a) \right)^2/2 = \delta^2(s,a)/2,$$

*where $\delta(s,a)$ is the Bellman error.*

Note that during a TS, at depth 1 we have access to $\delta(s_0, a)$ of all $a \in \mathcal{A}$ without additional computation. For depths 2 and above, the Bellman error is defined only for actions chosen by $\pi_o$, corresponding to a single trajectory down the tree. For these reasons, we base the above result on samples from depths 0 and 1.

Thanks to Prop. 3.6, we can estimate the bias correction term in Lemma 3.4 directly from the TS operation. Specifically, we substitute $\hat{\delta}_o/\sqrt{2}$ instead of $\sigma_o$ and the same for $\sigma_e$, where $\hat{\delta}_o$ is the Bellman error corresponding to $a = \pi_o(s)$ at the root, and $\hat{\delta}_e$ is the average Bellman error of all other actions. Hence, the correction term is

$$B(\hat{\delta}_e, \hat{\delta}_o, A, d) = \sqrt{\log A} \left( \hat{\delta}_e \sqrt{d} - \hat{\delta}_o \sqrt{d-1} \right) - (\hat{\delta}_e - \hat{\delta}_o)/\sqrt{8}. \tag{6}$$

A visualization of the resulting BCTS algorithm is in Fig. 2.

## 4 Solving Scalability via Batch-BFS

The second drawback of TS is scalability: exhaustive TS is impractical for non-trivial tree depths because of the exponential growth of the tree dimension. As TS requires generating $|A|^d$ leaves at depth $d$, it has been rarely considered as a viable solution. To mitigate this issue, we propose Batch-BFS, an efficient, parallel, TS scheme based on BFS, built upon the ability to advance multiple environments simultaneously; see Alg. 1. It achieves a significant runtime speed-up when a forward model is implemented on a GPU. Such GPU-based environments are becoming common nowadays because of their advantages (including parallelization and higher throughput) over their CPU counterparts. E.g., Isaac-Gym [37] provides a GPU implementation robotic manipulation tasks [46], whereas Atari-CuLE [11] is a CUDA-based version of the AtariGym benchmarks [4]. Batch-BFS is not limited to exact simulators and can be applied to learned deep forward models, like those in [36, 23]. Since Batch-BFS simultaneously advances the entire tree, it enables exhaustive tree expansion to previously infeasible depths. It also allows access to the cumulative reward and estimated value over all the tree nodes. Such access to all future values paves a path to new types of algorithms for, e.g., risk reduction and early tree-pruning. Efficient pruning can be done by maintaining an index array of unpruned states which are updated with each pruning step. These indices are then used for tracing the optimal action at the root. We leave such directions to future work. In addition to Alg. 1, we also provide a visualization of Batch-BFS in Fig. 3.

### 4.1 Runtime experiments

To showcase the efficacy of Batch-BFS, we compare it to a CPU-based BFS and to non-parallel TS, i.e., Depth-First-Search (DFS). We measure the duration of an average single TS operation on two environments: Atari-CuLE [11] and a Deep NN (DNN) mimicking a learned forward model. The

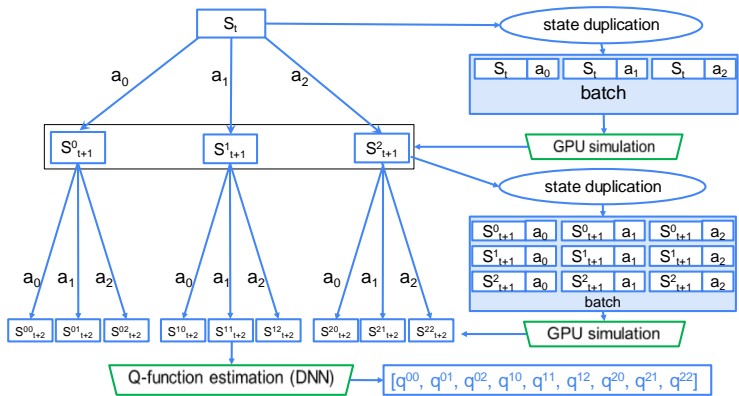

Figure 3: **Visualization of Batch-BFS.** The tree expansion is illustrated on the left, with the corresponding batch GPU operations on the right. In every tree expansion, the state $S_t$ is duplicated and concatenated with all possible actions. The resulting tensor is fed into the GPU forward model to generate the tensor of next states $(S_{t+1}^0, \ldots, S_{t+1}^{A-1})$. The next-state tensor is then duplicated and concatenated again with all possible actions, fed into the forward model, etc. This procedure is performed until the final depth is reached, in which case the Q-function is applied per state.

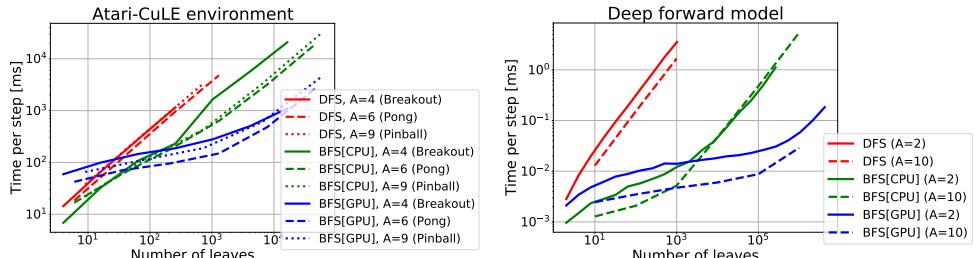

Figure 4: **Average tree-search time per action selection**. *Left:* Atari-CuLE Breakout, Pong, and VideoPinball. *Right:* A randomly generated neural network to mimic a learned forward-model with $A \in \{2, 10\}$. Note that $x$ and $y$ axes are in log-scale.

DNN is implemented in pytorch and uses cudnn [8]. It consists of three randomly-initialized hidden layers of width 100 with input size 100 for the state and 2 or 10 for the actions. The results are given in Fig. 4. We run our experiments on a 8 core Intel(R) Xeon(R) CPU E5-2698 v4 @ 2.20GHz equipped with one NVIDIA Tesla V100 16GB. Although we used a single GPU for our experiments, we expect a larger parallelization (and thus computational efficiency) to be potentially achieved in the future through a multi-GPU implementation. As expected, DFS scales exponentially in depth and is slower than BFS. When comparing BFS on CPU vs. GPU, we see that CPU is more efficient in low depths. This is indeed expected, as performance of GPUs without parallelization is inferior to that of CPUs. This issue is often addressed by distributing the simulators across a massive number of CPU cores [15]. We leverage this phenomenon in Batch-BFS by finding the optimal "cross-over depth" per game and swap the compute device in the middle of the search from CPU to GPU.

## 5   Experiments

In this section, we report our results on two sets of experiments: the first deals solely with TS for inference, without learning, whereas the second includes the case of TS used in training. In all Atari experiments, we use frame-stacking together with frame-skipping of 4 frames, as conducted in [31].

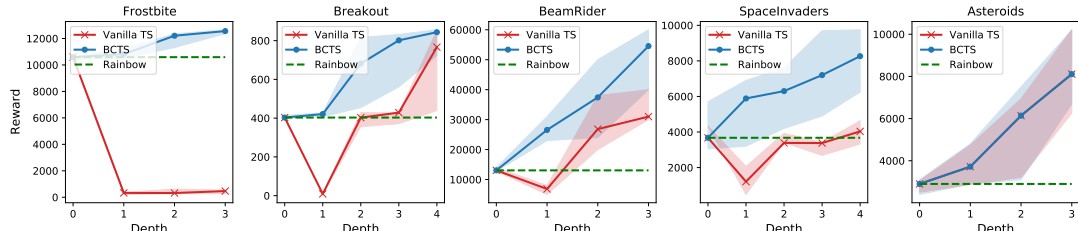

Figure 5: **Inference only: Vanilla TS vs. BCTS.** Median scores with lower (0.25) and upper (0.75) quantiles over 200 episodes, as a function of the tree depth. Surprisingly, vanilla TS often degrades the performance of the pretrained Rainbow agent. BCTS (blue) improves upon vanilla TS (red) for all depths, except for Asteroids. The improvement grows monotonically with the tree depth.

## 5.1 Inference with tree search

Using the pre-trained Rainbow agents in [24] (see Sec. 3), we test vanilla TS and show (Fig. 5, red plots) that it leads to a lower score than the baseline $\pi_o$. The largest drop is for $d = 1$ as supported by our analysis. The game that suffered the most is Frostbite. This can be explained from it having the largest number of actions ($A = 18$), which increases the bias in (5). As for BCTS, we found that multiplying its correction term (see (6)) by a constant that we sweep over can improve performance; we applied this method for the experiments here. Recall that BCTS is a TS applied on the pre-trained Rainbow baseline; i.e., the case of $d = 0$ is Rainbow itself. The results in Fig. 5 show that BCTS significantly improves the scores, monotonically in depth, in all games. It improves the Rainbow baseline already for $d = 1$, while for $d = 4$, the score more than doubles. For BeamRider, BCTS with $d = 4$ achieves roughly $\times 5$ improvement. Notice that without the computationally efficient implementation of Batch-BFS, the results for deeper trees would not have been obtainable in a practical time. We provide timing measurements per game and depth in Appendix B.1. Finally, notice that the advantage provided by BCTS is game-specific. Different games benefit from it by a different amount. In one of the games tested, Asteroids, vanilla TS was as useful as BCTS. Our findings reported in the last paragraph of Sec. 3.1 hint why certain games benefit from BCTS more than others. Nonetheless, a more thorough study on how the dynamics and state distribution in different games affect TS constitutes an interesting topic for future research.

## 5.2 Training with tree search

To further demonstrate the potential benefits of TS once a computationally efficient implementation (Section 4) is available, we show how it affects training agents from scratch on CuLE-Atari environments. We extend classic DQN [32] with TS using Batch-BFS for each action selection. Notice that training with TS does not suffer from the distribution shift studied in Sec. 3. Hence, the experiments below use vanilla TS and not BCTS.

Our experiment is even more significant considering that Efroni et al. [14] recently proved that the Bellman equation should be modified so that contraction is guaranteed for tree-based policies only when the value at the leaves is backed. However, this theory was not supported by empirical evidence beyond a toy maze. As far as we know, our work is the first to adopt this modified Bellman equation to obtain favorable results in state-of-the-art domains, thanks to the computationally efficient TS implementation achieved by Batch-BFS. We find this method to be beneficial in several of the games we tested. In the experiments below, we treat the Bellman modification from [14] as a hyper-parameter and include ablation studies of it in Appendix B.3.

We show the training scores in Table 1 and convergence plots in Appendix 9. For a fair comparison of different TS depths, we stop every run after two days on the same hardware (see Appendix D), not considering the total iteration count. To compare our results against classic DQN and Rainbow, we measure the number of iterations completed in two days by DQN with TS, $d = 0$. In Table 1 we report the corresponding intermediate results for DQN and Rainbow reported by the original authors in [40]. In most games, it amounts to roughly 30 million iterations. Note that DQN and Rainbow do not utilize the model of the environment while DQN with TS does; the former are brought as a reference for assessing the potential of using a TS with identical computation budget. As already shown in the case of inference with no training, the achieved score increases monotonically with

Table 1: **Atari scores after two days of training.** We follow the evaluation method in [22]: Average of 200 testing episodes, from the agent snapshot that obtained the highest score during training.

| Game | DQN with TS, depth $d$ | | | | DQN [32] | Rainbow [22] |
|---|---|---|---|---|---|---|
| | $d = 1$ | $d = 2$ | $d = 3$ | $d = 4$ | | |
| Asteroids | $2,093$ | $2,613$ | $4,794$ | $\mathbf{17,929}$ | $1,664$ | $1,594$ |
| Breakout | $385$ | $581$ | $420$ | $\mathbf{620}$ | $377$ | $327$ |
| MsPacman | $1,644$ | $2,923$ | $3,498$ | $\mathbf{4,021}$ | $2,398$ | $3,600$ |
| SpaceInvaders | $675$ | $1,602$ | $2,132$ | $\mathbf{2,550}$ | $1,132$ | $2,162$ |
| VideoPinball | $229,129$ | $244,052$ | $442,347$ | $345,742$ | $163,720$ | $\mathbf{641,235}$ |

the tree depth. In four of the five games, DQN with TS even surpasses the more advanced Rainbow. Since all results were obtained for identical runtime, improvement per unit of time is higher for higher depths. This essentially translates to better efficiency of compute resources. Convergence plots as a function of wall-clock time are shown in Appendix C. We also tested TS on two additional games not included in Table 1: Boxing and Pong. Interestingly, TS with $d = 4$ immediately obtained the highest possible scores in both these games already in the first training iteration.

## 6   Related work

The idea of searching forward in time has been employed extensively in control theory via methods such as A* [20], RRT [28], and MPC [2]. The latter is quite popular in the context of efficient planning in RL [36, 47, 48]. MPC-based controllers rely on recourse planning by solving an optimization program given the continuous structure of the dynamics. In our setup, the controls are discrete and the forward model is a black-box that cannot be directly used in an optimization scheme.

We leverage an efficient GPU simulator to conduct the look-ahead search. When such a simulator is out of reach, it can be learned from data. There is vast literature regarding learned deep forward models without optimizing a policy [26, 38, 27]. Other works [25, 23, 19, 41] used the learned model for planning but not with a TS. Few considered roll-outs of learned dynamics [6, 17] but only for evaluation purposes. Additional relevant works are MuZero [42] and "MuZero Unplugged" [43] which utilized a learned forward-model for prediction in an MCTS-based policy. In [34], the trade-off between learning and planning using a TS was empirically tested. Look-ahead policies for RL were also studied theoretically; bounds on the suboptimality of the learned policy were given in [12, 13, 14]. There, the focus was on the effect of planning on the learning process.

Finally, our distribution-shift analysis and approach draw connections to similar challenges in the off-policy [21, 18] and offline [29] RL literature. These address the issue of overestimation in RL due to the 'max' operation in the Bellman equation in various ways. In our work, we show that with TS the overestimation problem is exacerbated, and can lead to counter-intuitive performance degradation. Our analysis is different from previous papers, but intuitively the core solution is similar: "be careful" of states and actions not seen during training, and penalize them accordingly.

## 7   Discussion

Our study of the degradation with vanilla TS implies that the learned value function is not representative of the actual states that can be visited. This conclusion can be helpful in debugging RL systems. It can also be used to improve robustness, tune the approximation architecture, or guide exploration.

Our solution to the above performance degradation is an off-policy correction that penalizes high-error trajectories. It can be further improved with different notions of uncertainty for the value function, e.g., Bayesian or bootstrapped models. Also, while some simulators are available on GPU, such as Atari [11] and robotic manipulation [30], in other cases, a learned model can be used. In these cases, one could include the simulator quality in the off-policy penalty term. Finally, a limitation of our work is that we focus on problems with a discrete action space. Handling problems with continuous action tasks is a challenging direction for future work.

**Broader Impact.** The paper proposes a method to improve existing RL algorithms. As such, its main impact is to make RL more easily and widely deployable. Since our method can be applied to policies trained with any algorithm, it can be viewed as a generic "policy booster", and find applications with access to the environment model or its approximation.

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
