# A  Theory: Proofs and Complimentary Material

## A.1  Proof of Lemma 3.1

**Lemma.** *It holds that*

$$\hat{Q}_d^{\pi_o}(s, \pi_o(s)) = R_o(s) + \gamma^d G_o(s), \quad \max_{a \neq \pi_o(s)} \hat{Q}_d^{\pi_o}(s, a) = R_e(s) + \gamma^d G_e(s),$$

*with*

$$G_o(s) \sim GEV(\mu_o^{GEV}(s), \sigma_o^{GEV}, 0), \quad G_e(s) \sim GEV(\mu_e^{GEV}(s), \sigma_e^{GEV}, 0),$$

*where GEV is the Generalized Extreme Value distribution and*

$$\mu_o^{GEV}(s) = \mu_o(s) + \sigma_o \Phi^{-1}\left(1 - \frac{1}{A^{d-1}}\right),$$

$$\sigma_o^{GEV} = \sigma_o \left[\Phi^{-1}\left(1 - \frac{1}{eA^{d-1}}\right) - \Phi^{-1}\left(1 - \frac{1}{A^{d-1}}\right)\right],$$

$$\mu_e^{GEV}(s) = \mu_e(s) + \sigma_e \Phi^{-1}\left(1 - \frac{1}{A^d - A^{d-1}}\right),$$

$$\sigma_e^{GEV} = \sigma_e \left[\Phi^{-1}\left(1 - \frac{1}{e(A^d - A^{d-1})}\right) - \Phi^{-1}\left(1 - \frac{1}{A^d - A^{d-1}}\right)\right]. \tag{7}$$

*The function $\Phi^{-1}$ is the inverse of the CDF of the standard normal distribution.*

*Proof.* For $1 \leq i \leq A^{d-1}$, let $N_o^{(i)}$ be independent random variables distributed $N_o^{(i)} \sim \mathcal{N}(\mu_o(s), \sigma_o^2)$. According to (1), we have that

$$\hat{Q}_d^{\pi_o}(s, a) = \left[\max_{(a_k)_{k=1}^d \in \mathcal{A}} \left[\sum_{t=0}^{d-1} \gamma^t r(s_t, a_t)\right] + \gamma^d \hat{Q}^{\pi_o}(s_d, a_d)\right]_{s_0 = s, a_0 = a}.$$

For the case of $a = \pi_o(s)$, using Assumption 2, we can replace the cumulative reward above with $R_o(s)$ and be left with

$$\hat{Q}_d^{\pi_o}(s, \pi_o(s)) = R_o(s) + \gamma^d \max_{(a_k)_{k=1}^d \in \mathcal{A}} \hat{Q}^{\pi_o}(s_d, a_d)\big|_{s_0 = s, a_0 = \pi_o(s)}$$

$$= R_o(s) + \gamma^d \max_{1 \leq i \leq A^{d-1}} N_o^{(i)}$$

$$= R_o(s) + \gamma^d G_o(s),$$

where the second relation is due to Assumptions 1 together with 2, and the third follows from the maximum of $\{N_o^{(i)}\}_{i=1}^{A^{d-1}}$ being GEV-distributed with the parameters in the statement, derived as in [9]. Similarly, for $a_0 \neq \pi_0(s)$, we will have $A^d - A^{d-1}$ iid variables $N_e^{(i)} \sim \mathcal{N}(\mu_e(s), \sigma_e^2)$ and, consequently, $G_e(s)$ as given in the statement. $\square$

## A.2  Proof of Lemma 3.2

**Lemma.** *It holds that*

$$\mathbb{E}\left[\hat{Q}_d^{\pi_o}(s, \pi_o(s))\right] = Q_d^{\pi_o}(s, \pi_o(s)) + \gamma^d B_o(\sigma_o, A, d),$$

$$\mathbb{E}\left[\max_{a \neq \pi_o(s)} \hat{Q}_d^{\pi_o}(s, a)\right] = Q_d^{\pi_o}(s, a \neq \pi_o(s)) + \gamma^d B_e(\sigma_e, A, d),$$

*where the biases $B_o$ and $B_e$, satisfying $0 \leq B_o(\sigma_o, A, d) < B_e(\sigma_e, A, d)$, are given by*

$$B_o(\sigma_o, A, d) = \begin{cases} 0 & \text{if } d = 1, \\ \sigma_o \Phi^{-1}\left(1 - \frac{1}{A^{d-1}}\right) + \gamma_{EM}\sigma_o^{GEV} & \text{otherwise,} \end{cases}$$

$$B_e(\sigma_e, A, d) = \begin{cases} 0, & \text{if } d = 1 \text{ and } A = 2 \\ \sigma_e \Phi^{-1}\left(1 - \frac{1}{A^d - A^{d-1}}\right) + \gamma_{EM}\sigma_e^{GEV}, & \text{otherwise.} \end{cases} \tag{8}$$

*The constant $\gamma_{EM} \approx 0.58$ is the Euler–Mascheroni constant.*

*Proof.* First, obviously, the maximum over a set containing a single random variable has the distribution of that single element. Hence, there is no overestimation bias in the single-element case; i.e., the bias is 0 for the sub-tree of $a = \pi_o(s)$ with $d = 1$, and the sub-tree of $a \neq \pi_o(s)$ with $d = 1$ and $A = 2$.

Next, let $X$ be s.t. $X \sim \text{GEV}(\mu^{\text{GEV}}, \sigma^{\text{GEV}}, 0)$. From [9], we have that

$$\mathbb{E}\left[X\right] = \mu^{\text{GEV}} + \gamma_{\text{EM}}\sigma^{\text{GEV}}. \tag{9}$$

Applying Lemma 3.1, we have that

$$\mathbb{E}\left[\hat{Q}_d^{\pi_o}(s, \pi_o(s))\right] = \mathbb{E}\left[R_o(s) + \gamma^d G_o(s)\right]$$

$$= R_o(s) + \gamma^d \mathbb{E}\left[G_o(s)\right] \tag{10}$$

$$= R_o(s) + \gamma^d \left[\mu_o(s) + \sigma_o \Phi^{-1}\left(1 - \frac{1}{A^{d-1}}\right) + \gamma_{\text{EM}}\sigma_o^{\text{GEV}}\right] \tag{11}$$

$$= Q_d^{\pi_o}(s, \pi_o(s)) + \gamma^d B_o(\sigma_o, A, d), \tag{12}$$

where relation (10) is due to the rewards being deterministic (Assumption 2), relation (11) follows from Lemma 3.1 together with (9), and relation (12) is due to the definition of $Q_d^{\pi_o}(s, \pi_o(s))$ in (1), together with Assumptions 1 and 2. The calculation for the expectation $\mathbb{E}\left[\max_{a \neq \pi_o(s)} \hat{Q}_d^{\pi_o}(s, a)\right]$ follows the same steps.

Next, we show that

$$0 \leq B_o(\sigma_o, A, d) < B_e(\sigma_e, A, d). \tag{13}$$

For this, let us define

$$B(n) = \begin{cases} 0 & n = 1 \\ \gamma_{\text{EM}}\Phi^{-1}\left(1 - \frac{1}{en}\right) + (1 - \gamma_{\text{EM}})\Phi^{-1}\left(1 - \frac{1}{n}\right), & n \neq 1. \end{cases}$$

Notice we now have $B_o(\sigma_o, A, d) = \sigma_o B(A^{d-1})$ and $B_e(\sigma_e, A, d) = \sigma_e B(A^d - A^{d-1})$. Since by Assumption 1 we have that $\sigma_e > \sigma_o > 0$, to prove (13) it is sufficient to show that $0 \leq B(A^{d-1}) < B(A^d - A^{d-1})$. Since $B(n)$ is composed of two positive monotonically increasing functions, it is a positive monotonically increasing function. So, whenever $A^{d-1} < A^d - A^{d-1}$, we also have that $0 \leq B(A^{d-1}) < B(A^d - A^{d-1})$ and, consequently, (13). Taking a log, we see it is indeed the case for $A > 2$. For $A = 2$, we have equality and $B(A^{d-1}) = B(A^d - A^{d-1})$. But then, (13) holds again, since $\sigma_e > \sigma_o > 0$.

$\square$

### A.3  Proof of Theorem 3.3

**Theorem.** *The relation* $\mathbb{E}\left[\hat{Q}_d^{\text{BCTS}, \pi_o}(s, \pi_o(s))\right] > \mathbb{E}\left[\max_{a \neq \pi_o(s)} \hat{Q}_d^{\text{BCTS}, \pi_o}(s, a)\right]$, *holds if and only if* $Q_d^{\pi_o}(s, \pi_o(s)) > \max_{a \neq \pi_o(s)} Q_d^{\pi_o}(s, a)$.

*Proof.* For the case of $a = \pi_o(s)$, we have

$$\mathbb{E}\left[\hat{Q}_d^{\text{BCTS}, \pi_o}(s, \pi_o(s))\right] = \mathbb{E}\left[\hat{Q}_d^{\pi_o}(s, \pi_o(s))\right] = Q_d^{\pi_o}(s, \pi_o(s)) + \gamma^d B_o(\sigma_o, A, d),$$

where the first relation holds by the definition in (3) and the second relation follows from Lemma 3.2. Using the same arguments, for $a \neq \pi_o(s)$,

$$\mathbb{E}\left[\max_{a \neq \pi_o(s)} \hat{Q}_d^{\text{BCTS}, \pi_o}(s, a)\right]$$

$$= \mathbb{E}\left[\max_{a \neq \pi_o(s)} \hat{Q}_d^{\pi_o}(s, a)\right] - \gamma^d \left[B_e(\sigma_e, A, d) - B_o(\sigma_o, A, d)\right] + \gamma^d B_e(\sigma_e, A, d)$$

$$= \max_{a \neq \pi_o(s)} Q_d^{\pi_o}(s, a) + \gamma^d B_o(\sigma_o, A, d),$$

and the result immediately follows.

□

## A.4 Proof of Lemma 3.4

**Lemma.** *When $A^{d-1} \gg 1$, the correction term in (3) can be approximated with*

$$B_e(\sigma_e, A, d) - B_o(\sigma_o, A, d) \approx \sqrt{2 \log A} \left( \sigma_e \sqrt{d} - \sigma_o \sqrt{d-1} \right) - (\sigma_e - \sigma_o)/2. \qquad (14)$$

*Proof.* In the following, we apply the approximation

$$\Phi^{-1} \left( 1 - \frac{1}{n} \right) \approx \sqrt{2 \log n} - 0.5 \qquad (15)$$

from [3], which we empirically show to be highly accurate in Appendix A.7.

By definition (7),

$$\sigma_e^{\text{GEV}} = \sigma_e \left[ \Phi^{-1} \left( 1 - \frac{1}{e(A^d - A^{d-1})} \right) - \Phi^{-1} \left( 1 - \frac{1}{A^d - A^{d-1}} \right) \right]$$

$$\approx \sigma_e \left[ \Phi^{-1} \left( 1 - \frac{1}{eA^d} \right) - \Phi^{-1} \left( 1 - \frac{1}{A^d} \right) \right] \qquad (16)$$

$$\approx \sigma_e \left[ \sqrt{2 \log (eA^d)} - \sqrt{2 \log (A^d)} \right] \qquad (17)$$

$$= \sigma_e \frac{2 \log(eA^d) - 2 \log A^d}{\sqrt{2 \log (eA^d)} + \sqrt{2 \log A^d}}$$

$$\approx \frac{\sigma_e}{\sqrt{2d \log A}}, \qquad (18)$$

where in (16) we applied (15); in relation (17) we use that $A^d - A^{d-1} \approx A^d$ since $A^d \gg 1$; and in relation (18) we approximate $1 + \log(A^d) \approx \log(A^d)$, again because $A^d \gg 1$.

When $A^d \gg 1$, the second case of (8) holds, and thus

$$B_e(\sigma_e, A, d) = \sigma_e \Phi^{-1} \left( 1 - \frac{1}{A^d - A^{d-1}} \right) + \gamma_{\text{EM}} \sigma_e^{\text{GEV}}$$

$$\approx \sigma_e \left( \sqrt{2d \log A} - 0.5 \right) + \gamma_{\text{EM}} \frac{\sigma_e}{\sqrt{2d \log A}}$$

$$\approx \sigma_e \left( \sqrt{2d \log A} - 0.5 \right),$$

where the second relation follows from (15) and (18), while for the third relation we approximate $\sqrt{2 \log(A^d)}(1 + \frac{\gamma_{\text{EM}}}{2 \log(A^d)}) \approx \sqrt{2 \log(A^d)}$ since $A^d \gg 1$ (recall that $\gamma_{\text{EM}} \approx 0.58$).

Applying the same derivation for $B_o$ gives that

$$B_o(\sigma_o, A, d) \approx \sigma_o \left( \sqrt{2(d-1) \log A} - 0.5 \right), \qquad (19)$$

and the result follows directly. □

## A.5 Proof of Theorem 3.5

To prove Theorem 3.5, we first obtain the following non-approximate result.

**Theorem A.1.** *The policy $\pi_d^{BCTS}(s)$ (see (4)) chooses a sub-optimal action with probability bounded by:*

$$\Pr \left( \pi_d^{BCTS}(s) \notin \arg\max_a Q_d^{\pi_o}(s, a) \right) \leq \left( 1 + \frac{6 \left( Q_d^{\pi_o}(s, \pi_o(s)) - \max_{a \neq \pi_o(s)} Q_d^{\pi_o}(s, a) \right)^2}{\gamma^{2d} \pi^2 \left( (\sigma_o^{GEV})^2 + (\sigma_e^{GEV})^2 \right)} \right)^{-1}.$$

$$(20)$$

*Proof.* We first recall Cantelli's inequality [7]: Let $X$ be a real-valued random variable. Then, for some $\lambda > 0$,

$$\Pr\left(X - \mathbb{E}\left[X\right] \geq \lambda\right) \leq \left(1 + \frac{\lambda^2}{\mathrm{Var}[X]}\right)^{-1}. \tag{21}$$

We choose

$$X := \max_{a \neq \pi_o(s)} \hat{Q}_d^{BCTS,\pi_o}(s, a) - \hat{Q}_d^{BCTS,\pi_o}(s, \pi_o(s)).$$

We can now split the event of sub-optimal action choice as follows:

$$\{\pi_d^{\mathrm{BCTS}}(s) \notin \arg\max_a Q_d^{\pi_o}(s, a)\}$$
$$= \{\pi_o(s) \in \arg\max_a Q_d^{\pi_o}(s, a) \cap X > 0\} \bigcup \{\pi_o(s) \notin \arg\max_a Q_d^{\pi_o}(s, a) \cap X < 0\}. \tag{22}$$

Note that this division is possible because according to Assumptions 1 and 2, all actions different than $\pi_o(s)$ have equal cumulative reward and value in expectation.

Next, we consider the two (deterministic) following possible cases.
**Case I:** $\pi_o(s) \in \arg\max_a Q_d^{\pi_o}(s, a)$. Then, the second event in (22) is an empty set and we have that

$$\{\pi_d^{\mathrm{BCTS}}(s) \notin \arg\max_a Q_d^{\pi_o}(s, a)\} = \{X > 0\}. \tag{23}$$

**Case II:** $\pi_o(s) \notin \arg\max_a Q_d^{\pi_o}(s, a)$. Then, from symmetry, we will get $\{\pi_d^{\mathrm{BCTS}}(s) \notin \arg\max_a Q_d^{\pi_o}(s, a)\} = \{X < 0\}$.

In the rest of the proof, we shall apply Cantelli's inequality to upper bound $P(X > 0)$ in Case I, i.e., to bound the sub-optimal action selection event. Afterward, we will explain how Case II yields the same bound as in Case I.

Let us set

$$\lambda = -\mathbb{E}\left[X\right] = Q_d^{\pi_o}(s, \pi_o(s)) - \max_{a \neq \pi_o(s)} Q_d^{\pi_o}(s, a), \tag{24}$$

where the second relation follows from the proof of Theorem 3.3. Next, we calculate the variance of $X$ using GEV theory [9]. For $G \sim \mathrm{GEV}(\mu, \sigma, 0)$, we have that $\mathrm{Var}\left[G\right] = \sigma^2 \frac{\pi^2}{6}$. Then,

$$\mathrm{Var}\left[X\right] = \mathrm{Var}\left[\max_{a \neq \pi(s)} \hat{Q}_d^{BCTS,\pi_o}(s, a) - \hat{Q}_d^{BCTS,\pi_o}(s, \pi_o(s))\right]$$
$$= \mathrm{Var}\left[\max_{a \neq \pi(s)} \hat{Q}_d^{BCTS,\pi_o}(s, a)\right] + \mathrm{Var}\left[\hat{Q}_d^{BCTS,\pi_o}(s, \pi_o(s))\right]$$
$$= \mathrm{Var}\left[\max_{a \neq \pi(s)} \hat{Q}_d^{\pi_o}(s, a)\right] + \mathrm{Var}\left[\hat{Q}_d^{\pi_o}(s, \pi_o(s))\right]$$
$$= \frac{\gamma^{2d} \left(\sigma_e^{\mathrm{GEV}}\right)^2 \pi^2}{6} + \frac{\gamma^{2d} \left(\sigma_o^{\mathrm{GEV}}\right)^2 \pi^2}{6}, \tag{25}$$

where the third relation is because $\mathrm{Var}\left[\hat{Q}_d^{BCTS,\pi_o}(s, a)\right] = \mathrm{Var}\left[\hat{Q}_d^{\pi_o}(s, a)\right]$ following (3); the last relation follows from $\hat{Q}_d^{\pi_o}$ having a GEV distribution as given in Lemma 3.1.

Plugging (23), (24), and (25) into (21) gives the desired result for Case I. Finally, for Case II, we define $Y = -X$ and repeat exactly the same process to upper bound $P(Y > 0)$. Since $(\mathbb{E}[Y])^2 = (\mathbb{E}[X])^2$, and $\mathrm{Var}[Y] = \mathrm{Var}[X]$, we obtain exactly the same bound as in Case I. This concludes the proof. $\square$

Theorem 3.5 now follows from Theorem A.1 after plugging the approximation (18) from the proof of Lemma 3.4 in (20), and upper bounding the resulting expressions:

$$\sigma_e^{\mathrm{GEV}} \approx \frac{\sigma_e}{\sqrt{2d \log A}} \leq \frac{\sigma_e}{\sqrt{d \log A}}, \qquad \sigma_o^{\mathrm{GEV}} \approx \frac{\sigma_o}{\sqrt{2(d-1) \log A}} \leq \frac{\sigma_o}{\sqrt{d \log A}}.$$

## A.6  Proof of Proposition 3.6

**Proposition.** *Let $\widehat{var}_n[X]$ be the variance estimator based on $n$ samples of X. Then,*

$$\widehat{var}_{n=2}[\hat{Q}^{\pi_o}(s,a)] = \left(\hat{Q}_1^{\pi_o}(s,a) - \hat{Q}_0^{\pi_o}(s,a)\right)^2/2 = \delta^2(s,a)/2,$$

*where $\delta(s,a)$ is the Bellman error.*

*Proof.* We shall employ the known unbiased variance estimator:

$$\widehat{var}_{n=2}[\hat{Q}^{\pi_o}(s,a)]$$

$$= \frac{1}{n-1} \sum_{d=0}^{n} \left(\hat{Q}_d^{\pi_o}(s,a) - \frac{1}{n}\sum_{d=0}^{n} \hat{Q}_d^{\pi_o}(s,a)\right)^2$$

$$\overset{(n=2)}{=} \left(\hat{Q}_0^{\pi_o}(s,a) - \frac{\hat{Q}_0^{\pi_o}(s,a) + \hat{Q}_1^{\pi_o}(s,a)}{2}\right)^2 + \left(\hat{Q}_1^{\pi_o}(s,a) - \frac{\hat{Q}_0^{\pi_o}(s,a) + \hat{Q}_1^{\pi_o}(s,a)}{2}\right)^2$$

$$= \frac{1}{4}\left(\hat{Q}_0^{\pi_o}(s,a) - \hat{Q}_1^{\pi_o}(s,a)\right)^2 + \frac{1}{4}\left(\hat{Q}_1^{\pi_o}(s,a) - \hat{Q}_0^{\pi_o}(s,a)\right)^2$$

$$= \frac{\delta^2(s,a)}{2}$$

$\square$

## A.7  Approximate bounds bias

To show the validity of our approximation, we numerically evaluate the two sides of (5), i.e. the LHS

$$B_e(\sigma_e, A, d) - B_o(\sigma_o, A, d)$$

as given in (8), vs. the RHS

$$\sqrt{2\log A}\left(\sigma_e\sqrt{d} - \sigma_o\sqrt{d-1}\right) - (\sigma_e - \sigma_o)/2.$$

To compute the values, we arbitrarily choose $A = 3, \sigma_o = 1, \sigma_e = 4$, and increase $d$. We plot the results in Fig. 6. As seen, the approximation is highly accurate for all values of $A^d$.

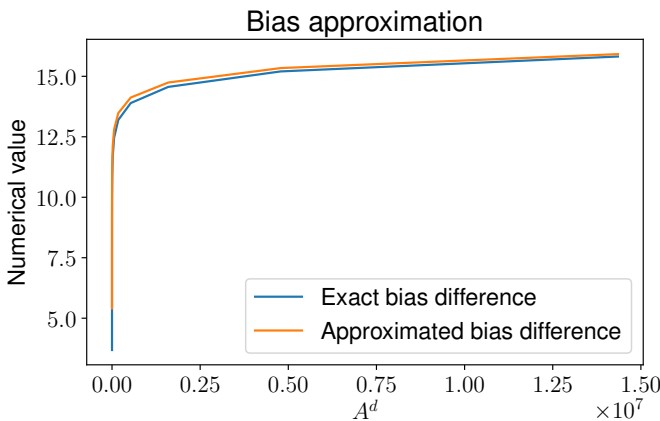

Figure 6: **Quality of approximation in Lemma 3.4.**

# B  Experiments complementary material

## B.1  Inference timing

We measure the average complete TS time per action selection. We provide the results together with the respective scores per game and depth in Fig. 7. The scores are obtained via BCTS with a Batch-BFS implementation, as reported in Section 5.1. As depth increases, the number of explored nodes

in the tree grows exponentially, with runtime increasing accordingly. The plots depict the tradeoff between improved scores and the corresponding price in terms of runtime per action selection.

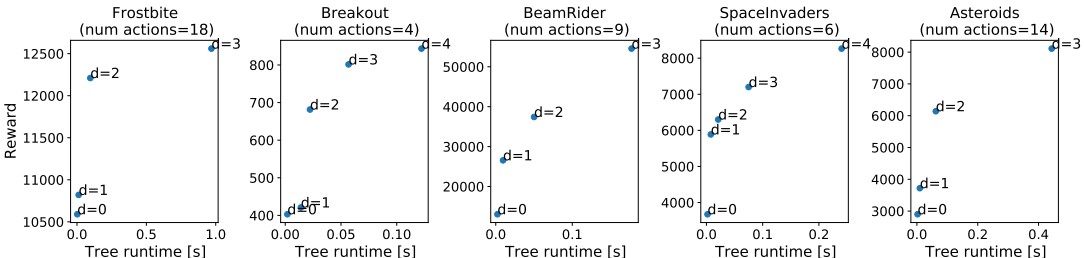

Figure 7: **Score vs. inference time of each TS operation as a function of TS depth.**

## B.2 Training

In our training experiments we use the same training hyper-parameters as the original DQN paper [32].

## B.3 Ablation study: Training with propagated value from the tree nodes

Here, we present an ablation study for the correction to the Bellman update proposed in [14] in the case of a TS policy. This correction modifies the training target: Instead of bootstrapping the value from the transition sampled from the replay buffer, we use the cumulative reward and value computed during the TS. In Fig. 8, we present training plots for Atari Space-Invaders for DQN with TS of depths 2,3, and 4. Note that for depth 1, the correction is vacuous since it coincides with the classic Bellman update. As seen, for Space-Invaders, the correction improves convergence in all tested depths.

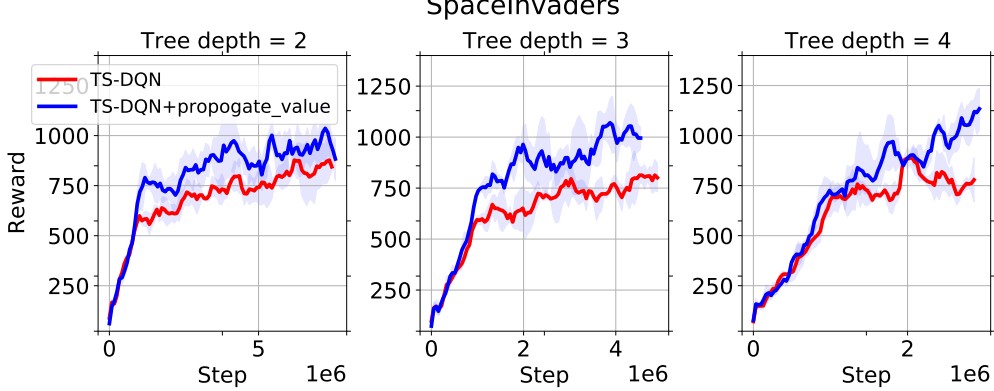

Figure 8: **Propagated value from the tree nodes: Ablation study for Space-Invaders convergence.** Episodic training cumulative reward of DQN with TS based on 5 seeds. We compare the standard update method with the update based on the propagated value from the tree nodes, as proposed in [14].

Lastly, we summarize the results for all tested games in Table 2. The table reveals that the correction often improves training, though not always. Therefore, we treat it as a hyper-parameter that we sweep over.

## C   Training: Wall-clock time

We provide in Fig. 9 convergence plots of DQN with TS for all tested Atari games, for depths 0 to 4. To showcase the time efficiency of using a tree-based policy, we give the scores with respect to

Table 2: **Ablation study: Propagated value (PV) from the tree nodes: Ablation study for scores of all tested games.** All agents of all depths were trained using DQN with TS, with similar train time that amounts to 30 million frames of DQN. The reported scores were obtained as in [22]: Average of 200 testing episodes, from the agent snapshot that obtained the highest score during training.

| Game | $d = 2$ | | $d = 3$ | | $d = 4$ | |
|---|---|---|---|---|---|---|
| | PV=False | PV=True | PV=False | PV=True | PV=False | PV=True |
| Asteroids | $2,613$ | $2,472$ | $4,794$ | $4,693$ | $17,929$ | $15,434$ |
| Breakout | $568$ | $581$ | $420$ | $417$ | $620$ | $573$ |
| MsPacman | $2,514$ | $2,923$ | $3,498$ | $3,046$ | $2,748$ | $4,021$ |
| SpaceInvaders | $1,314$ | $1,602$ | $1,204$ | $2,132$ | $1,452$ | $2,550$ |
| VideoPinball | $214,168$ | $244,052$ | $442,347$ | $366,670$ | $345,742$ | $301,752$ |

the wall-clock time. We run each training experiment for two days, which amount to roughly 30 million frames for depth 0 and 1 million frames for depth 4. For each run, we display the average score together with std using 5 seeds.

Fig. 9 reveals the trade-off between deeper TS inference (action selection) time and the improvement thanks to the deeper TS. For regular DQN (depth=0), each inference is the fastest, but generally leads to lower scores than deeper trees. On the other hand, the deepest tree is not necessarily the most time-efficient. In most cases here, there is a sweet-spot in depth 2 or 3 that gives the best score for the same training time.

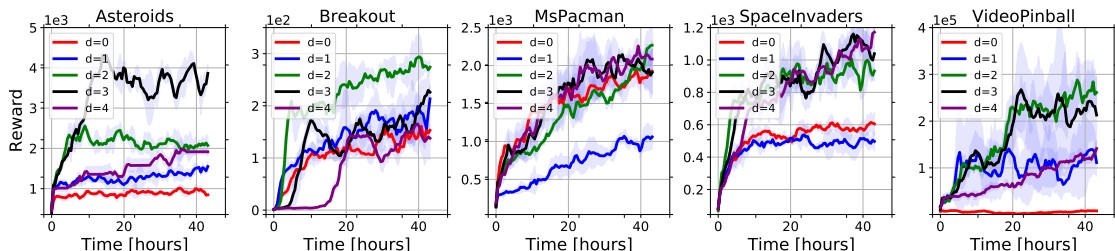

Figure 9: **Training convergence plots for tree search with DQN of depths up to** $4$**.**

## D   Hardware

We run our training experiments on a 10 core Intel(R) CPU i9-10900X @ 3.70GHz equipped with NVIDIA Quadro GV100 32GB.