# OpenReview forum: "Improve Agents without Retraining: Parallel Tree Search with Off-Policy Correction"
_NeurIPS.cc/2021/Conference — NeurIPS 2021 Poster_

### Official Review · Reviewer_drZi · 2021-07-13

**Rating:** 7
**Confidence:** 5

**Summary:**

The authors introduce a new tree search technique that can be used to improve the score of trained RL agents online. They propose an analysis of the counter-intuitive phenomenon that applying standard MCTS with a pre-trained value function is less performant than using directly the value function in spite having access to more information.

The first contribution is an analysis of TS and the introduction of an off-policy correction term based on the analysis. The second contribution is a GPU adaptation of TS making it fast at inference time but also possible to train tree based agents in a scalable way.

The paper is well written and addresses an interesting problem and the proposed method could be used in many RL algorithms and applications. The GPU implementation seems somewhat disconnected to the rest of the paper.


**Ethical Concerns:**

No particular concerns.

**Limitations And Societal Impact:**

This work requires models or simulators for tree search to be applied but overall it can touch a broad range of applications beyond games.

**Main Review:**

The authors attribute the performance loss of standard tree search to a distribution shift. When using the estimated value function to guide the tree search, it is evaluated in states poorly explored during training, with inaccurate value estimate and hence downgrade the performance of the whole tree search.

Empirical analysis of the degradation caused by TS:

They provide empirical proof and a further theoretical analysis. I think the empirical proof lack of details, fortunately I had observed this phenomena in my own experiments and was not to be convinced.
-	“We also measure the off-policy distribution shift between \pi_0 and the TS policy by counting disagreements “ In this above sentence, do the author use pi_0 in the TS as well?
-	This section on empirical proof of the discrepancy between TS and pi_0 lacks a bit of precision. The author should provide more details at least in appendix. Does the computation budget of the tree search affect these results? Did they use any exploration policy (e.g. UCB) in the tree search?

Regarding the theoretical analysis:

The author assume that the value estimate follows a normal distribution with two different deviations depending on whether the action follows the original policy or not. Although simplistic this assumption seems reasonable enough to provide a first analysis of the problem and is well justified by the authors.

In assumption 2: shouldn’t the value function be Q_d^{\pi_0} instead of Q^{\pi_0}? The justification for assumption 2 is not as clear.
Throughout the analysis the author separate the cases where the action follows the original policy or other branches of the tree. The dichotomy is used in all the subsequent proofs. It results in two different biases for the “pi-0 following branch” and the other branch.

The two biases are used to provide a correction term in the action selection procedure of the tree. With another approximation, the authors derive a very simple formula for approximating the correction term as a function of: the size of the action space, the depth of the tree, sigma_e, sigma_o. The two standard deviations are introduced in assumption 1.

The rest of the analysis shows that the probability of selecting a suboptimal action is bounded.

The overall analysis strongly rely on assumption 1 but seems sound and correct overall, leading to a practical correction term with two parameters to estimate. The remaining of the section provide guidance in how to estimate sigma_o and sigma_e. The author should provide an algorithm, or an intuitive explanation on how to practically estimate those two parameters, I think proposition 3.6 provides a theoretical justification but it is unclear how to practically implement it.

Scalability:

This part addresses a completely different problem of TS: scalability. TS is usually expensive to run as it requires many online simulations growing exponentially with the depth. The authors rely on GPUs to address the scalability issue. This contribution is not theoretical but practical.

The author provide a pseudo-code of the solution but no explanation is given. The diagram in the appendix is helpful and should in my opinion replace the provided pseudo-code.

Without surprise, the parallel implementation outperforms standard TS for a large number of leaves in terms of runtime. The batch BFS implementation consists in evaluating each level of the tree at the same time in a “batch”. It is unclear how this method can be used with TS techniques using pruning or UCB type branching.

Inference Experiments:

The experiments support the claim, BCTS provides consistent improvement over rainbow and standard TS.
“As for BCTS, we found that multiplying its correction term (see (6)) by a constant that we sweep over can improve performance; we applied this method for the experiments here.”
This constant is not mentioned in the analysis and should be further discussed. If sweeping of a constant is being used, one wonders if estimating delta_e and delta_o is really useful or if we could just do parameter sweeping as well.

Training experiments:

The author don’t use the BCTS algorithm but the batch-BFS only for this set of experiments. However the author mention that they use another corrected update? Clarification needed.
The results in table 1 should also include the pre-trained rainbow + BCTS for a full comparison of the proposed methods.


**Time Spent Reviewing:**

1.5

---

> ### Author Response · Authors · 2021-08-09
> **Thank you for your review. Here are our answers to all questions raised.**
>
> (1) **” In this above sentence, do the author use $\pi_0$ in the TS as well?”**
> Yes, we use the value of $\pi_0$ in the TS and compare it to directly using $\pi_0$ without TS altogether.
>
> (2) **”This section on empirical proof of the discrepancy between TS and $\pi_0$ lacks a bit of precision. The author should provide more details at least in appendix. Does the computation budget of the tree search affect these results? Did they use any exploration policy (e.g. UCB) in the tree search?”**
> More computational budget would amount to a deeper search. In these experiments we describe the agreement for depth 1. Those agreements vary as depth increases in a non-monotone way. We shall provide the results for all inspected depths in the appendix. Second, since the TS is conducted exhaustively for multiple states in parallel, no exploration is needed.
>
> (3) **”In assumption 2: shouldn’t the value function be $Q_d^{\pi_o}$ instead of $Q^{\pi_o}$?”**
> Making the assumption on the value $Q_d^{\pi_o}$ at the root is also possible, but it would have been stronger than the assumption we currently make. Instead, we only assume the value $Q^{\pi_o}$ at the leaf itself is a Gaussian RV, and later show that it causes the value at the root $Q_d^{\pi_o}$ to be GEV-distributed, which causes the overestimation effect.
>
> (4) **”The justification for assumption 2 is not as clear.”**
> In the general case, every trajectory will have different cumulative rewards and value at the leaf. Comparing only the highest estimated Q-functions will therefore include only a subset of the leaves. In our analysis, we wanted to find the worst case bias, which is obtained when all trajectories behave the same in terms of cumulative rewards and value at the leaf. This is at the base of Assumption 2 -- there is no additional signal to separate between the $A^d$ leaves besides the initial action. Naturally this also makes the analysis simpler as less parameters need to be defined.
>
> (5) **”The remaining of the section provide guidance in how to estimate $\sigma_o$ and $\sigma_e$. The author should provide an algorithm, or an intuitive explanation on how to practically estimate those two parameters. I think proposition 3.6 provides a theoretical justification but it is unclear how to practically implement it.”**
> We will make this point clearer in the text before Eq. (6). Following Proposition 3.6, we simply use the square of the Bellman error divided by 2 for estimating $\sigma_o$ and $\sigma_e$, where the Bellman error is immediately obtainable from the TS. Plugging this into Lemma 3.4, we obtain Eq. (6).
>
> (6) **”It is unclear how this method can be used with TS techniques using pruning or UCB type branching.”**
> This can be done by maintaining an index array of unpruned states which are updated with each pruning step. They are finally used for tracing the optimal action at the root. We did not mention this technical point as it was not used in the paper; we will add this to the appendix.
>
> (7) **”This constant is not mentioned in the analysis and should be further discussed.”**
> The benefit of Eq. (6) with $\delta_o$ and $\delta_e$ is the per-state computation. Replacing it with a single constant leads to poor performance as different states should have different corrections. So a parameter sweep could not replace the estimation of $\delta_o$ and $\delta_e$. Instead, we only scale the per-state computation with a constant.  We could have included this coefficient in the analysis, but this would have made the expressions more complex while adding very little value.
>
>
> (8) **"The author don’t use the BCTS algorithm but the batch-BFS only for this set of experiments. However the author mention that they use another corrected update? Clarification needed.“**
> As written in Section 5.2, the corrected update for the Bellman loss utilizes the cumulative reward computed by the TS and adds it to the NN Q-function at the leaf for a more accurate value estimation. This correction was proposed and experimented with in [14], but the experiments were conducted on toy problems. We will clarify this better in the text.
>
> (9) **”The results in table 1 should also include the pre-trained rainbow + BCTS for a full comparison of the proposed methods.”**
> Good point, we will add this column.

---

> > ### Comment · Reviewer_drZi · 2021-08-31
> > **Thank you for answering my questions**
> >
> > The answers were very clear and helped.
> > I still maintain my choice to accept this paper, I hope the author will release an open source implementation.

---

### Official Review · Reviewer_1GjW · 2021-07-13

**Rating:** 7
**Confidence:** 3

**Summary:**

This paper is about combinations of Tree Search (TS) and trained value functions, where the value functions may already have been trained without involving any TS during training, in RL problems such as Atari games. Claimed contributions:

1) Identification and analysis of a distribution-shift phenomenon, where, if a value function was trained standalone (without TS), adding TS on top of it at evaluation time can actually reduce an agent's performance (relative to, say, a plain agent that is greedy w.r.t. the trained value function). The basic intuition for why this happens is as follows: during training, states that are more likely to be visited under the greedy policy (w.r.t. value function) show up more frequently in collected experience, and hence also get a more accurately-trained value function. A tree search (especially an exhaustive one) will run the value function significantly more often to also evaluate states that the greedy policy would rarely/never reach, and in these states the value function will have larger errors (which may be overestimating or underestimating errors). Because there are usually more paths through the tree that the greedy policy would not visit than there are paths that the greedy policy would visit, the number of branches with large errors are likely to outnumber the branches with smaller errors. This outnumbering leads to a maximisation bias, where, when there are multiple such branches with large errors, likely at least one of them will be overestimating, causing the tree search to prioritise that branch over one with a more accurately-trained value function.

2) A correction term that allows a new "BCTS" tree search algorithm to function well with a pre-trained value function.

3) An approach named Batch-BFS to efficiently, on a GPU, run a breadth-first-style tree search with trained value function (assumes, I think, that the forward model of the environment, like an Atari game, itself can also be implemented to run on the GPU).

The resulting Batch-BFS approach outperforms DQN and Rainbow on several Atari games.

**Limitations And Societal Impact:**

Limitations are adequately addressed in the paper, except possibly somewhat of a limitation in the experiments, see point 3) in the main review above.

**Main Review:**

Overall I think the paper is solid in all the core aspects (originality, quality, clarity, and significance). I do have several more detailed comments below, but do not think any of them should be a major issue to address:

1) First two paragraphs of Introduction discuss how the paradigm of MCTS would be less suitable for continuous/large state spaces or image-based domains, and how the more simple "breadth-first-style" TS used in this paper would be more suitable without requiring any memorization, but I don't see why this is true. MCTS has no more problems with continous or large state spaces than a BFS would. With an extremely large or continuous state space, MCTS would likely struggle to grow a sufficiently large tree to fully cover that space... but that holds for any other tree search too. I wonder, with the references to image-based domains and memorization, are the authors actually thinking of problems with a large state **representation** (rather than state space), i.e. problems where a single state requires a large amount of memory to store? Again, I'm not sure that I agree that this is necessarily that much of a problem for MCTS. If it takes too much memory to stores states in nodes, it is also possible to implement MCTS by only storing actions along the edges, and use those actions to recreate the state as the algorithm traverses the tree.

2) Section 4 states that Batch-BFS is proposed "to solve this issue" [= the issue that the number of nodes to evaluate scales exponentially with search depth d]. I think that "solve" may be too strong a word here. I'll happily believe that GPUs allow for significantly more parallelisation than CPUs, and can meaningfully increase the search depth that can feasibly be reached... but I don't see that as really **solving** the issue of exponential growth. There is still exponential growth, and exponential growth will at some point catch up again and even make a GPU struggle. Maybe just something like "mitigate" or "address" would be more fair. On a related note, it would be interesting to see what kinds of search depths match the numbers of leaves reported on the x-axes of Figure 3.

3) In *"Revisiting the Arcade Learning Environment: Evaluation Protocols and Open Problems for General Agents"*, Machado et al. (2018) investigated how, due to the determinisim in ALE, many approaches turn out to be able to perform well in these environments simply by memorising strong action sequences, and propose using *sticky actions* (in training as well as evaluation) to disincentivise such solutions (as opposed to ones that actually learn to make informed decisions based on the state and generalise to new circumstances). Were sticky actions used in any of the trainings or evaluations in this paper? I think this is a particularly interesting question in the context of this paper, because the use of sticky actions during training would force a bit more exploration and force the value function to be applicable to a more diverse set of states than in a (close to) deterministic environment. To what extent would such a training practice mitigate the bias and maybe reduce or eliminate the need for a correction?

---

Minor comments:
- Quotes on left-hand side of words "Left" and "Right" on bottom of page 3 should "point" the other way (opening instead of closing quotes)
- Footnote 1 states the Atari environments used are "close to deterministic". Is it possible to elaborate on in what way(s) they are *not* deterministic?
- Line 123: evidence support --> evidence supports
- Line 154: Technically, I guess it does not necessarily hold exactly that the expectation of Q-hat equals Q when trained using Rainbow? There may be errors due to imperfect function approximation, probably a bias from Prioritized Experience Replay since it's customary to only partially correct for its non-uniform sampling with importance sampling, and if I recall correctly also only partial or no correcting for biases due to multi-step off-policy update steps in Rainbow. Maybe all of this is not too relevant to this paper though.
- In Figure 2, is it correct to state that the red states were "not visited/observed during training"? I get that they are likely visited *less frequently*, but not entirely unvisited, right?
- Caption Figure 4: "a deeper tree often degrades the performance of vanilla TS" --> is it really fair to say "often" here? I see this primarily being the case just in Frostbite, plus in most of the other games specifically when going from depth 0 --> 1, but not anymore when increasing depth beyond 1.

**Time Spent Reviewing:**

2.5

---

> ### Author Response · Authors · 2021-08-09
> **Thank you for your review. Here are our answers to all questions raised.**
>
> (1) **Summary of contribution list**
> We are happy that the reviewer took the time to explain in great accuracy the details and essence of our work and appreciates it.
>
> (2) **”I'm not sure that I agree that this is necessarily that much of a problem for MCTS. If it takes too much memory to stores states in nodes, it is also possible to implement MCTS by only storing actions along the edges, and use those actions to recreate the state as the algorithm traverses the tree.”**
> We first recall that MCTS works by gradually expanding a memorized tree using some UCB-type exploration technique, until hitting a leaf for which value/rollouts are computed. When the MDP is deterministic and always begins at the same initial state (e.g., Go), indeed the reviewer’s suggestion for recreating trajectories using sequence of actions would work well. However, in the case of stochastic dynamics (e.g., autonomous vehicles (AVs)), MCTS has the inherent problem of recreation: taking the same sequence of actions may lead to different states, which would not follow the memorized MCTS tree. A non-memorized “on-demand” TS, on the other hand, does not rely on the assumption that the same action sequence would lead to the same tree traversal, and tackling stochasticity can be done simply via multiple sampling (for simplicity of analysis and experiments, we do focus on deterministic environments in this work). Second, if the initial state is drawn randomly (again, AVs in different road scenarios) -- the specific initial state has to either exist and be found in the MCTS tree or alternatively a new tree has to be built and reiterated for that initial state, which is impractical. The above problems can be mitigated in tabular MDPs or effectively small-sized via discretization, but it is not clear how to do so for large/continuous state-spaces. We will clarify the above points better in the paper.
>
> (3) **”Section 4 states that Batch-BFS is proposed "to solve this issue" ... I think that "solve" may be too strong a word here.”**
> Fair point, we agree. We will replace “solve” with “mitigate”.
>
> (4) **”On a related note, it would be interesting to see what kinds of search depths match the numbers of leaves reported on the x-axes of Figure 3.”**
> We will add these numbers as an additional x-axis.
>
> (5) **”Were sticky actions used in any of the trainings or evaluations in this paper? ... the use of sticky actions during training would force a bit more exploration and force the value function to be applicable to a more diverse set of states than in a (close to) deterministic environment. To what extent would such a training practice mitigate the bias and maybe reduce or eliminate the need for a correction?”**
> We indeed use four steps of sticky actions together with frame-stacking for each node expansion. So did the agents that were used to train the pre-trained value functions. We’ll clarify it in the experiments section. We agree that without sticky actions, the bias problem would likely impair the results even more since the overfitting effect would be more intense. Notice, also, that a standard practice during training is to gradually anneal epsilon (as in epsilon-greedy) towards some low minimal value (common defaults: epsilon=0.1 achieved after 0.1 of the total steps). Following the reviewer’s line of thought, using a higher lower bound could potentially help mitigate the bias problem.
>
>
>
> (6) **”Footnote 1 states the Atari environments used are "close to deterministic". Is it possible to elaborate on in what way(s) they are not deterministic?”**
> There are some environments which include stochasticity in the transitions like the ghost movement in MsPacman. We shall specify that in the footnote.
>
> (7) **”Line 154: Technically, I guess it does not necessarily hold exactly that the expectation of Q-hat equals Q when trained using Rainbow?”**
> We assume the Rainbow SGD converges to some point in the proximity of a “true” value function. There are many factors influencing this random training process, such as those you listed. If we repeat the process with a different seed, we will converge to some other point in that neighborhood. However, we don’t have a reason to believe that for a given state, these points of convergence should be mostly above or below the “true” value of that state. This explains why we assume no bias of the SGD convergence point. Note also that this is a pure SGD regression question and is not related to the overestimation effect discussed in the paper.
>
> (8) **”In Figure 2, is it correct to state that the red states were "not visited/observed during training"? I get that they are likely visited less frequently, but not entirely unvisited, right?”**
> Correct, we will refine the text.
>
> (9) **”Caption Figure 4: "a deeper tree often degrades the performance of vanilla TS" --> is it really fair to say "often" here?“**
> Correct, we will rephrase to be more accurate.
>
> **”Quotes on left-hand side of words "Left" and "Right"”; “evidence support --> evidence supports“**
> (10) Thank you; we will fix these typos.

---

> > ### Comment · Reviewer_1GjW · 2021-08-25
> > **Just a small note on MCTS in stochastic environments**
> >
> > Thanks for your responses!
> >
> > I just have one small note in response to the second point, about MCTS and issues it would run into when not storing states in nodes in stochastic environments. The authors may be interested in the paper "Open Loop Search for General Video Game Playing" by D. Perez et al. (2015). They describe a variant of MCTS very much alike the one I suggested, in that it does not store states in nodes but instead re-generates them every time the tree is traversed... and this was actually proposed specifically for stochastic games -- because in this way it allows multiple possible "futures" to be simulated and sampled from, rather than just one.
> >
> > I do believe this would lose the guarantee we normally expect from MCTS that it converges to optimal solutions given infinite time, but I suppose the same would hold for a similar multiple-sampling variant (without explicitly accounting for stochasticity through chance nodes) of the breadth-first-style TS.
> >
> > This is really not a major concern or anything like that for my review, just a small note that I figured the authors may find useful or interesting.

---

### Official Review · Reviewer_AYhs · 2021-07-16

**Rating:** 6
**Confidence:** 2

**Summary:**

In the setting where a pretrained value function is used in combination with MCTS, the authors show that because of the distribution shift problem, using the pretrained value function as the evaluation function in MCTS results in a performance worse than directly using the value function. Then, a correction term that in the end is based on the bellman error is proposed to resolve this issue. Experiments show that this correction term is effective.

The second part of this paper proposes a more GPU compatible search method that seems to scale well when a GPU simulator is available.

**Limitations And Societal Impact:**

My main suggestion is to improve the presentation of this submission, especially Section 3.

Second, I think more experimental results on other domains will also make the submission stronger. I am also interested to see search time constrained evaluation to understand to what extend the correction term can allow the performance of MCTS to be improeved with a pretrained value function.

Minor:
- is there link to be made to [1]?


[1] Jiang, Nan, et al. "The dependence of effective planning horizon on model accuracy." Proceedings of the 2015 International Conference on Autonomous Agents and Multiagent Systems. 2015.

**Main Review:**

Clarity:
- the clarity of Section 3 can be greatly improeved. Can some of the lemmas be moved to the appendix?
- also, I think more intuitive explaination of the correction term can help readers better understand the main idea. I am feeling that it is an intuitive method in the end that has been overcomplicated.

Originality:
- the presented methods are new as I far as I know.

Quality:
- overall this submission seems technically sound.
- what are the limitations of your methods?
- does assumption 1 really holds in practice?

Significance:
- this submission presents a nice solution to an interesting problem, which is the distribution shift problem when using pretrained value function in MCTS.

**Time Spent Reviewing:**

6

---

> ### Author Response · Authors · 2021-08-09
> **Thank you for your review. Here are our answers to all questions raised.**
>
> **”(1) The clarity of Section 3 can be greatly improved. Can some of the lemmas be moved to the appendix?”**
> We shall move Lemma 3.1, which is the more technical of the two, to the appendix. Lemma 3.2 is important for introducing the results thereafter, so we will keep it in a simpler form.
>
> (2) **”I think a more intuitive explanation of the correction term can help readers better understand the main idea.”**
> This is a good point. The method may seem complicated due to the formality of the paper. But, at the end it is indeed very intuitive: the idea is to distinguish between actions that are truly “good”, and those that are “within the range of noise”. So, by quantifying the noise using the Bellman error, we manage to find the formula as a function of the parameters (Bellman-error, $d, A, \gamma$) that gives what exact debiasing would yield the optimal signal-to-noise separation. We shall add this intuition at the beginning of Sec. 3.
>
> (3) **”what are the limitations of your methods?”**
> TS requires some kind of forward model (l. 47). To use BatchBFS efficiently, a GPU-based forward model is required (l. 233). Lastly, the analytical expression of the correction term is based on Assumptions 1 and 2.
>
> (4) **”does assumption 1 really holds in practice?”**
> Assumption 1 consists of two parts: (a) Gaussian distribution for $Q^{\pi_o}$ and (b) higher variance for values corresponding to $\pi_o$ vs the rest of the actions. We indeed validate (b) at the end of Subsection 3.1. However, (a) does not necessarily hold in practice. But, the role of (a) is only to simplify the bias expression in Lemma 3.2 (application of the GEV). One can derive his own GEV expression using other distributions in a plug-and-play fashion.
>
>
> (5) **”Second, I think more experimental results on other domains will also make the submission stronger.”**
> Additional domains would require efficient simulators, preferably in GPU. We recently added such an implementation for simple tasks such as cartpole. We will include those simulations in the appendix of the updated copy.
>
> (6) **”I am also interested to see search time constrained evaluation to understand to what extend the correction term can allow the performance of MCTS to be improved with a pretrained value function.”**
> Thank you for the suggestion, we will run such an experiment and add a corresponding plot.
>
> (7) **”Is there a link to be made to [1]?”**
> Thank you for the reference; we will add it to the paper. The use-case of [1] is when an inaccurate learned forward model is used instead of an exact simulator. In that case, TS with lower depths could lead to better results since relying on multiple-step state prediction would incur high error.

---

### Official Review · Reviewer_z7iw · 2021-07-17

**Rating:** 6
**Confidence:** 3

**Summary:**

This work investigates the problem of improving pretrained agents on discrete RL problems by using tree search. Using a ground-truth environment model, it finds that naively implemented tree search sometimes results in worse performance than no tree search at all. The authors provide analysis of this phenomenon in a simplified setting which shows that performing search on noisy Q estimates, especially with greater noise for states outside the span of the policy, results in taking incorrect actions. They introduce a correction term for tree search and show that it provides superior performance. The paper also includes a discussion of using batching to make tree search faster on accelerators, including using it to train agents.

**Limitations And Societal Impact:**

Yes

**Main Review:**

This work has some interesting elements, but is currently a bit unpolished and is not ready for publication. The paper would benefit from increased focus on the discussion of overestimation and its correction, the most interesting and novel components. The section on batching forward passes during search for efficiency is described as novel but is routine in the model-based RL community. The discussion of training with tree search distracts from the central topic of the paper as it does not involve the problems of overestimation and distribution shift.

### Structure and text
The basic setting, using a learned agent at test time with a greater degree of tree search than was used at training time, is straightforward and fairly common in model-based RL; see for example the appendix of [5]. The introduction would be stronger if it did not re-introduce this idea before discussing the contribution of this work specifically.

Figure 2 is nice and would be helpful if moved into the introduction to help set the scene for the reader.

In Figure 1, I believe the Q-values shown are the estimates (i.e. $\hat Q$). It would be helpful if the \hats were applied more consistently throughout the work.

Lemmas 3.1 and 3.2 are extremely unhelpful as written since they consist entirely of terms defined in the appendix. It would be much better if they were at least stated in a self-contained way, though of course the proofs can be deferred to the appendix. Even replacing them with easy-to-understand bounds would be more illuminating, especially given that we have already made some strong assumptions so the precise details are not so important.

`\widehat` would be helpful for defining the variance in Proposition 3.6.

While the top-line numbers for BCTS are good, this work would be much more compelling with additional investigative experiments. Is overestimation on little-visited states happening in practice during tree search? How much of the time? Does that over-estimation decay as predictions are further in the future, or does performance improve with increasing $d$ due to the $\gamma$ decay alone? (This would be surprising since $\gamma$ is usually very near 1.) Does the overestimation scale as predicted by your theory? Overall the theory is fine, but the assumptions are such that it should only be thought of as something to build intuition; I want to know if that's _really_ what's going on with a learned agent.


### Technical questions
In equation (1), I believe the max should be applied to both terms, i.e. the max over actions of (accumulated reward + bootstrap Q value)?

Assumption 1 deals with state and action trajectories, and defines one variance if the first action matches the policy and a different one otherwise. What about the rest of the actions? That is, as the assumption is written now, the variance of $\hat Q$ would be low even if every action other than the first was different than the one $\pi_o$ would take. Similarly in Assumption 2. Is this intended, and necessary to simplify the analysis?

Theorem 3.3 is maybe not quite what one would want to prove — the expected value of the estimator might be correct even while it produces incorrect actions much of the time. Indeed, by the assumptions we already have that $\hat Q$ is unbiased, and yet optimizing against it has bad outcomes. It might be difficult to prove, but a more compelling result would be about exactly how often this algorithm takes the optimal action with respect to the true $Q_d^{\pi_0}$.

Do you use sticky actions, as described by [4]? This makes the environments substantially less deterministic.

The results shown in Table 1 on training with tree search are striking, but the comparisons are not necessarily the most meaningful. Training with tree search is a question of policy optimization against a  _known model_, whereas the full reinforcement learning setting (which the baselines are operating in) involves no prior knowledge about the environment. I'm not too bothered with this though since I would recommend removing this section entirely.


### Novelty
This examination of overestimation and distribution shift is interesting, but not compellingly novel. I have seen related ideas in the literature though probably not constructed in exactly this way.

This paper touches on an interesting fundamental question that is fundamental in off-policy and offline reinforcement learning: how well does a Q function generalize, and how can we mitigate its variance for policy optimization? In the off-policy literature, there might be useful connections to draw to [2] and [3]. For offline RL, [1] might make a good starting point in the literature.

### References
1. Regularized Behavior Value Estimation, Gulcehre 2021
2. Double Q-learning, van Hasselt 2010
3. Addressing Function Approximation Error in Actor-Critic Methods, Fujimoto 2018
4. Revisiting the Arcade Learning Environment: Evaluation Protocols and Open Problems for General Agents, Machado 2017
5. Mastering Atari, Go, Chess and Shogi by Planning with a Learned Model, Schrittwieser 2019

**Time Spent Reviewing:**

5

---

> ### Author Response · Authors · 2021-08-09
> **Thank you for your review. Here are our answers to all questions raised.**
>
> (1) **”The section on batching forward passes during search for efficiency is described as novel but is routine in the model-based RL community.”**
> Previous works (e.g., [5]) used TS with a learned model. Our novelty is in showing that the combination of batch forward-pass together with the structure of BFS is very efficient. The contribution of creating an exact-simulator TS on GPU for atari is new and can serve others. Furthermore, our runtime studies in Fig. 3 are new and can be beneficial for researchers in the field.
>
> (2) **”The discussion of training with tree search distracts from the central topic of the paper as it does not involve the problems of overestimation and distribution shift.”**
> Training with TS is an additional contribution that follows easily with our method. While it is not the central part of the paper, it gives a quantitative baseline of how much a simple DQN agent can be improved by adding only TS to it and an estimate of its computational cost (as we show -- not high when using GPU simulation). We believe other researchers can benefit from our novel observation that applying TS of non-trivial depth is feasible with millions of training steps and, as Fig. 9 shows, allows improving sample complexity.
>
> (3) **”see for example the appendix of [5]. The introduction would be stronger if it did not re-introduce this idea before discussing the contribution of this work specifically.”**
> The learned model in [5] is used for MCTS, and as such has two key differences wrt our work. First, MCTS is different from the exhaustive search that we use (see first two paragraphs of introduction). With MCTS and any other TS that is used for training, there is no distribution shift of chosen actions. Hence, MCTS does not give rise to the phenomenon we analyze in our work. Second, in the appendix of [5] (latest arxiv version) we could not find any mention of using a different tree search depth in training and in evaluation. It’s also not clear how such different depths can be used with MCTS: evaluation uses the last tree from training, so they are of the same depth. We will clarify these two differences in the final version.
>
> (4) **”Figure 2 is nice and would be helpful if moved into the introduction to help set the scene for the reader.”**
> Good idea, we will incorporate it in the introduction.
>
> (5) **”In Figure 1, I believe the Q-values shown are the estimates (i.e. Q^). It would be helpful if the \hats were applied more consistently throughout the work.”**
> You are correct, we will update the notation.
>
> (6) **”Lemmas 3.1 and 3.2 are extremely unhelpful as written since they consist entirely of terms defined in the appendix.”**
> Following this suggestion, we will move details to the appendix while keeping the core message in the main text. Specifically, we will move Lemma 3.1 to the appendix, and replace it with a description of its key point: The Q estimates are distributed according to a known “max” distribution with computable parameters. The main point of Lemma 3.2 is that this distribution can be used to directly compute the bias which is an analytical expression of $\sigma_e, \sigma_o , A, d$, and $\gamma$, and that the bias is higher for the non-native actions. We will keep Lemma 3.2 in the main paper, but revise the details to make it more accessible.
>
> (7) **”widehat would be helpful for defining the variance in Proposition 3.6.”**
> Thank you, we will update it.
>
> (8) **” While the top-line numbers for BCTS are good, this work would be much more compelling with additional investigative experiments. Is overestimation on little-visited states happening in practice during tree search? How much of the time?”**
> We included these experiments in Subsection 3.1. For completeness, we summarize them here. The experiment in l.129 shows that the agreement between the original and tree-based policy is only 18% for Breakout and 2% for Frostbite.  Those low agreement values suggest that overestimation occurs often for Breakout, and more often for Frostbite. They explain the performance degradation in the red curve of Fig 4, which is worst for Frostbite. Not every disagreement is due to an overestimation, but the fact that the pretrained policy is highly performing (fully trained Rainbow) together with the low agreement levels suggests that the vast majority are. Nonetheless, it is impossible to quantify exactly how much of the time there is actual overestimation and how it depends on d. To measure the bias at a given state, one should subtract its estimated value $\hat{Q}^{\pi_o}_d(s, \pi_o(s))$ from its true value $Q^{\pi_o}_d(s, \pi_o(s))$, to which we don’t have access. Lastly, note that in addition to this experiment, in l.126 we also validated the basis to Assumption 1 by showing that the Bellman error (which, as Prop 3.6 shows, is an estimation of the variance) of the branch corresponding to $\pi_o$ is significantly smaller than that corresponding to the other actions.
>
> (9) **”In equation (1), I believe the max should be applied to both terms, i.e. the max over actions of (accumulated reward + bootstrap Q value)?”**
> Good catch, we will fix this typo.
>
> (10) **”Assumption 1 deals with state and action trajectories, and defines one variance if the first action matches the policy and a different one otherwise. What about the rest of the actions?”**
> To simplify the theory we indeed assume the variance and cumulative rewards that correspond to every leaf in the subtree depend only on the first action, even if all future actions are different from $\pi_o$. It is possible to perform a more careful analysis where the variance depends on every state-action along each trajectory in the tree, but that would make the theory more complicated (with additional variances and rewards defined per trajectory) while the essence is the same. That kind of analysis would lead to per-leaf correction instead of a single correction for the entire subtree.
>
> (11) **”Theorem 3.3 is maybe not quite what one would want to prove… It might be difficult to prove, but a more compelling result would be about exactly how often this algorithm takes the optimal action with respect to the true $Q_d^{\pi_o}$.”**
> Theorem 3.5 gives exactly the requested result. It bounds the probability that our algorithm selects a suboptimal action w.r.t. the true $Q_d^{\pi_o}$. As for Theorem 3.3, notice that it shows that the BCTS correction manages to fix the bias.
>
> (12) **”Do you use sticky actions, as described by [4]? This makes the environments substantially less deterministic.”**
> Indeed, we use four steps of sticky actions together with frame-stacking for each node expansion. We will clarify it in the experiments section.
>
> (13) **”The results shown in Table 1 on training with tree search are striking, but the comparisons are not necessarily the most meaningful.”**
> We agree with this point and will add a disclaimer stating it. Nevertheless, this comparison is important for demonstrating how much the “shallow” algorithms can benefit from adding a TS to them. Also, in the appendix, Fig. 9, we provide a convergence plot as a function of runtime that compares depths 0 to 4 to show that given a simulator, a sample complexity optimum can be achieved for some non-zero depth.
>
> (14) **”This examination of overestimation and distribution shift is interesting, but not compellingly novel.”**
> As noted by the reviewer, TS is a canonical tool in RL. Thus, the novelty and significance in our paper are three folds:  First, in discovering how a naive TS may impair performance. Second, in quantifying the detrimental effect of overestimation. Third, in how we quantify the overestimation. As far as we know, this is the first paper using GEV theory for that purpose; we believe our analysis is simple and intuitive compared to the standard RL theoretical tools.
>
> (15) **”In the off-policy literature, there might be useful connections to draw to [2] and [3]. For offline RL, [1] might make a good starting point in the literature.”**
> We thank the reviewer for these references and will include them in the literature review. All of [1-3], as well as the recent series of DICE papers for off-policy/offline RL [6], address the issue of overestimation in RL due to the ‘max’ operation in the Bellman equation in various ways. In our work, we show that with TS the overestimation problem is exacerbated , and can lead to counter-intuitive performance degradation. Our analysis and solution are different from previous papers, but intuitively the core idea is similar: “be careful” of states/actions not seen during training, and (in some of the references) penalize them accordingly.
> [6] Lee, Jongmin, et al. "OptiDICE: Offline Policy Optimization via Stationary Distribution Correction Estimation." arXiv preprint arXiv:2106.10783 (2021).
> ---
> Lastly, hoping the reviewer finds our answers satisfactory, we would kindly ask the reviewer to consider raising his score.

---

> > ### Comment · Reviewer_z7iw · 2021-08-25
> > **Reviewer response**
> >
> > Thanks for the responses.
> >
> > ### Inline replies
> >
> > > (3) With MCTS and any other TS that is used for training, there is no distribution shift of chosen actions
> >
> > This is only true when the exact same tree search hyperparameters (e.g. depth, ponder time) are used in both train and test; prior work has explored increasing the ponder time during testing.
> >
> > > (3) Second, in the appendix of [5] (latest arxiv version) we could not find any mention of using a different tree search depth in training and in evaluation. It’s also not clear how such different depths can be used with MCTS
> >
> > This is shown in Figure S3 of the appendix of [5] on Arxiv: https://arxiv.org/pdf/1911.08265.pdf
> >
> > > (8) The experiment in l.129 shows that the agreement between the original and tree-based policy is only 18% for Breakout and 2% for Frostbite. Those low agreement values suggest that overestimation occurs often for Breakout, and more often for Frostbite.
> >
> > Not precisely. These experiments only show that these two policies choose different actions. That could be due to systematic overestimation on little-visited actions as suggested, but could also be due to any other kind of value estimation errors (including variance rather than bias).
> >
> > > (8) Nonetheless, it is impossible to quantify exactly how much of the time there is actual overestimation and how it depends on d. To measure the bias at a given state, one should subtract its estimated value from its true value, to which we don’t have access
> >
> > Given that you have a fast, parallel GPU implementation of Atari, the common solution to estimating the true value would be to snapshot the environment and compute a Monte Carlo estimate of the true value using rollouts.
> >
> > > (11) Theorem 3.5 gives exactly the requested result
> >
> > You are correct and I apologize for the oversight.
> >
> > ### Overall
> >
> > Given the authors' replies as well as the observations of the other reviewers, I will raise my score.

---

### Author Response · Authors · 2021-08-09
**We wish to thank the reviewers.**

We thank the reviewers for spending the time and effort to carefully evaluate our work. The reviewers found our work **“solid in all the core aspects (originality, quality, clarity, and significance)” [R 1GjW]**, **"well written"** and addressing **"an interesting problem and the proposed method could be used in many RL algorithms and applications” [R drZi]**. They also found it  **"technically sound”**, including **"methods that are new"**, and representing **“a nice solution to an interesting problem” [R AYhs]**. Beyond these encouraging descriptions, the reviewers also made valuable comments that we answered point to point in the following.

---

> ### Author Response · Authors · 2021-09-02
> **Follow-up**
>
> We thank the reviewers for reading and considering our clarifications and hope we have answered all major points.

---

### Decision · Program_Chairs · 2021-09-27

**Decision:**

Accept (Poster)

**Comment:**

This is a solid paper and set of contributions. There are certainly areas where it could be improved; to me these are primarily around the structure of the paper and writing (not that these are poor, but there are these two essentially distinct contributions that some reviewers get tripped up on). z7iw points out that related ideas can be found in the literature and that this negatively impacts novelty, and I agree that this overlap exists but would argue that whatever minor hit novelty takes it is made up for by how this can unify understanding of where these challenges arise. There are several concrete suggestions (or areas for clarification) that the reviewers have brought up, and the authors are encouraged to include them in the final version of the paper.